

**Change in Frozen Soils and its Effect on Regional Hydrology in the**
**Upper Heihe Basin, the Northeast Qinghai-Tibetan Plateau**
Bing Gao[1], Dawen Yang[2]*, Yue Qin[2], Yuhan Wang[2], Hongyi Li[3], Yanlin Zhang[3], and
Tingjun Zhang[4]
[1] School of Water Resources and Environment, China University of Geosciences,
Beijing 100083, China
[2] State Key Laboratory of Hydroscience and Engineering, Department of Hydraulic
Engineering, Tsinghua University, Beijing 100084, China
[3] Cold and Arid Regions Environmental and Engineering Research Institute, Chinese
Academy of Sciences, Lanzhou, Gansu 730000, China
[4] Key Laboratory of West China's Environmental Systems (MOE), College of Earth
and Environmental Sciences, Lanzhou University, Lanzhou, 730000, China
*Correspondence to*:   Dawen Yang (yangdw@tsinghua.edu.cn)





**ABSTRACT:**
Frozen ground has an important role in regional hydrological cycle and ecosystem,
especially on the Qinghai-Tibetan Plateau, which is characterized by high elevation and
a dry climate. This study modified a distributed physically-based hydrological model
and applied it to simulate the long-term (from 1961 to 2013) change of frozen ground
and its effect on hydrology in the upper Heihe basin located at Northeast Qinghai-
Tibetan Plateau. The model was validated carefully against data obtained from multiple
ground-based observations. The model results showed that the permafrost area shrank
by 9.5% (approximately 600 $km^2$), especially in areas with elevation between 3500 m
and 3900 m. The maximum frozen depth of seasonally frozen ground decreased at a
rate of approximately 4.1cm/10yr, and the active layer depth over the permafrost
increased by about 2.2 cm/10yr. Runoff increased significantly during cold seasons
(November-March) due to the increase in liquid soil moisture caused by rising soil
temperature. Areas where permafrost changed into the seasonally frozen ground at high
elevation showed especially large changes in runoff. Annual runoff increased due to
increased precipitation, the base flow increased due to permafrost degradation, and the
actual evapotranspiration increased significantly due to increased precipitation and soil
warming. The groundwater storage showed an increasing trend, which indicated that
the groundwater recharge was enhanced due to the degradation of permafrost in the
study area.
**KEYWORDS:** permafrost; frozen ground; active layer, soil moisture; soil temperature;
runoff, distributed hydrological model



## 1. Introduction

Hydrological processes on the Qinghai-Tibetan Plateau, which is characterized by high
elevation and cold climate are greatly influenced by cryosphere processes. In recent
years, the runoff change in the Qinghai-Tibetan Plateau has received increasing
attentions due to its significant effect on water resources and the ecosystem (Cuo et al.,
2014). The change in frozen soils and its effect on hydrological processes is a key
scientific issue (Yang et al., 2010; Cheng and Jin, 2013). Frozen soils including
permafrost and seasonally frozen ground, have active interactions with land surface
hydrological processes. Changes in frozen soils alter land surface infiltration, soil
drainage, and subsurface water storage and influences the partition of direct surface
runoff and subsurface flow. Hydrological changes caused by frozen soils can greatly
impact land-atmosphere interactions and thus the water balance and energy balance of
the land surface. Understanding the changing frozen soil conditions and their impact on
hydrological processes is important for water resources management and ecosystem
protection on the Qinghai-Tibetan Plateau.
Several previous observation-based studies have examined long-term changes in
frozen soils and their impacts on hydrological processes. Some studies reported that
permafrost thawing might enhance base flow, especially runoff in winter in the Arctic
and the Subarctic (Walvoord et al., 2016; Jacques and Sauchyn, 2009; Ye et al., 2009)
and in Northeast China (Liu et al., 2003). A few studies argued that permafrost thawing
might reduce river runoff (Qiu, 2012). These studies used either in-situ observations in
experimental catchments or long-term meteorological observations. Field experiments



are usually at the plot scale for a short period, which might lose the spatial variability
and long-term trends, and the long-term meteorological observations do not provide
data on soil freezing and thawing processes (McClelland et al., 2004; Liu et al., 2003;
Niu et al., 2011). Previous observation-based studies focus either on runoff trends or
changes in frozen soils; few studies thoroughly discuss the relationship between runoff
trends and changes in frozen soils. The impact of the change in frozen soils on regional
hydrological processes is not fully understood based on the existing observations and it
is difficult to attribute the long-term trends of streamflow to the change in frozen soils
(Woo et al., 2008).

Hydrological models have been widely used to analyze the regional hydrological

changes under changing environmental conditions; however most hydrological models
do not consider the freezing-thawing processes in soil. Some studies incorporate simple
freezing-thawing schemes into the hydrological models (Rawlins et al., 2003; Chen et
al., 2008), but do not simulate the soil thermal fluxes. The SiB2 model (Sellers et al.,
1996), the modified VIC model (Cherkauer and Lettenmaier, 1999) and the CLM model
(Oleson et al., 2010) consider the land surface energy balance and soil heat transfer
processes, but do not represent the complex landscape at the catchment scale. The
GEOtop model simulates the three-dimensional water flux and vertical heat transfer in
soil, but it is difficult to apply at the regional scale. Wang et al. (2010) and Zhang et al.
(2013) incorporated frozen soil schemes in a distributed hydrological model and
showed improved performance in a small mountainous catchment. Rawlins et al. (2013)
analyzed the impact of future climate change at 4 sites in Alaska. Subin et al. (2013)



and Lawrence et al. (2015) used the CLM model to simulate the change in permafrost
at global scale. Cuo et al. (2015) simulated frozen soil degradation and its effects on
surface hydrology at the plot scale using the VIC model. The previous modelling
studies focused on simulations of the changes in frozen soils and the hydrological
impacts at either the small scale or global/continental scale. Regional modelling studies
linking the frozen soils changes and hydrological responses were inadequate.

The Qinghai-Tibetan Plateau is the Asian water tower, and water availability on the

plateau is very important for water supply and food security in the downstream regions
with large populations (Walter et al., 2010). Different from the Arctic and Subarctic,
the permafrost thickness on the Qinghai-Tibetan Plateau is relatively thin and warm,
and the frozen depth of the seasonally frozen soils is also shallow (Yang et al., 2010).
Therefore, frozen soil processes in the Qinghai-Tibetan Plateau are more sensitive to
rising air temperature (Yang et al., 2010). Due to the drier climate and warmer soil, the
frozen soil processes are more closely related to the hydrological processes on the
Qinghai-Tibetan Plateau than they are in the Arctic and Subarctic regions. There is also
higher spatial variability in topography and landscapes on the Qinghai-Tibetan Plateau
where the permafrost and seasonally frozen ground coexist.
An evident increase in the annual and seasonal air temperature has been observed
in the Qinghai-Tibetan Plateau (Li et al., 2005; Liu and Chen, 2000; Zhao et al., 2004).
Several studies have shown changes in frozen soils based on the long-term observations.
For example, Cheng and Wu (2007) analyzed the borehole observations of soil
temperature profiles on the Qinghai-Tibetan Plateau and found that the active layer



thickness of frozen soils increased by 0.15-0.50 m during the period of 1996-2001.
Zhao et al. (2004) found a decreasing trend of freezing depth in the seasonal frozen
soils using observations from 50 stations. Several studies analyzed the relationship
between the change in frozen soils and streamflow based on observed data (Zhang et
al., 2003; Jin et al., 2009; Niu et al., 2011). However, these studies have not addressed
the spatial and temporal variations of the frozen soils. The spatio-temporal
characteristics of the long-term change in frozen soils is not sufficiently clear. The
magnitude of the effect of frozen soils on regional hydrology remains unclear, and the
modelling studies on frozen soils changes and their hydrological impacts are
insufficient. Therefore, integrated study based on the long term simulation of soil
freezing/thawing processes and the hydrological responses is needed.

Through a comprehensive experiment (Li et al., 2013) in a major research plan

entitled "Integrated research on the ecohydrological processes of the Heihe basin"
funded by the National Natural Science Foundation of China (NSFC) (Cheng et al.,
2014), this study aims: (1) to develop a distributed hydrological model coupling the
cryosphere processes especially the soil freezing-thawing processes; (2) to simulate the
spatial and temporal changes in frozen soils and to analyze the effects of frozen soils
change on hydrological processes in the upper Heihe basin located on the Northeastern
Qinghai-Tibetan Plateau.
**2.  Study area and data**
**2.1 The Heihe River and upper Heihe basin**
The Heihe River is one of the major inland basins in Northwest China. As shown in



Figure 1, the upper reaches of Heihe River are located on the Northeastern Qinghai-
Tibetan Plateau at an elevation of 2200-5000 m and with a drainage area of 10009 km$^2$,
it supplies most of the water resources to the middle and lower reach (Cheng et al.,
2014). The annual precipitation in the upper Heihe basin ranges from 200 to 700 mm,
and the annual mean air temperature ranges from -9 to 5℃. Permafrost dominates high
elevation region above 3700 m (Wang et al., 2013) and seasonal frozen ground covers
other parts of the study area. Glaciers are found at an elevation above 4000 m, covering
approximately 0.8% of the upper Heihe basin. There are two tributaries (East and West
Tributaries) in the upper Heihe basin, on which two hydrological stations are located,
namely, Qilian (on the east tributary) and Zhamashike (on the west tributary). The outlet
of the upper Heihe basin has a hydrological station, namely Yingluoxia (see Figure 1).
**2.2 Data used in the study**
**(1) Input data of the model**
The atmosphere forcing data used to drive the hydrological model include a 1-km
resolution gridded dataset of daily precipitation, air temperature, sunshine hours, wind
speed and relative humidity. The gridded daily precipitation is interpolated from
observations at meteorological stations (see Figure 1) provided by the China
Meteorological Administration (CMA) using the method developed by Shen and Xiong
(2015). The other atmosphere forcing data are interpolated by observations at
meteorological stations using the inverse distance weighted method. The interpolation
of air temperature considers the temperature gradient with elevation which is provided
by the HiWATER experiment (Li et al., 2013).




The land surface data used to build the model include land use, topography, leaf
area index, and soil parameters. The topography data are obtained from the SRTM
dataset (Jarvis et al., 2008) with a spatial resolution of 90 m. The land use/cover data
are provided by the Institute of Botany, Chinese Academy of Sciences (Zhou and Zheng,
2014). The leaf area index (LAI) data with 1-km resolution are obtained from the
dataset developed by Fan (2014). The soil water parameters and soil physical
parameters of each grid are obtained from the 1-km dataset developed by Song et al.
(2016), which includes the saturated hydraulic conductivity, residual soil moisture
content, saturated soil moisture content, soil sand matter content, soil clay matter
content and soil organic matter content.
**(2) Data used for model calibration and validation**
This study uses the observed daily river discharge data at the Yingluoxia, Qilian
and Zhamashike stations, the daily soil temperature of different depths at the Qilian
station and the daily frozen depths at the Qilian and Yeniugou stations for model
calibration and validation. Daily river discharge data are obtained from the Hydrology
and Water Resources Bureau of Gansu Province. Daily soil temperature data observed
at the Qilian station which is from January 1, 2004 to December 31, 2013 and daily
frozen depth data observed at the Qilian and Yeniugou station from January 1, 2002 to
December 31, 2013 are provided by CMA.
To investigate the spatial distribution of permafrost, boreholes were drilled during
the NSFC major research plan. Temperature observations at six boreholes, whose
location are shown in Figure 1, are provided by Wang et al. (2013). The borehole depth



is 100 m for T1, 69 m for T2, 50 m for T3 and 90 m for T4, and 20 m for T5 and T7.
Monthly actual evapotranspiration data with 1-km resolution during the period of 2002-
2012 estimated using remote sensing data (Wu et al., 2012; Wu, 2013) are used to
evaluate the model-simulated evapotranspiration. We also used field observations of
the hourly liquid soil moisture to validate the model simulation of frozen soils. The
HiWATER experiment (Li et al., 2013; Liu et al., 2011) provided the soil moisture data
observed at the A'rou Sunny Slope station (100.52 E, 38.09 N), which is available from
January 1, 2014 to December 31, 2014.
**3.    Methodology**
**3.1 Brief introduction of the hydrological model**
This study used a distributed hydrological model GBEHM (geomorphology-based
ecohydrological model), which was developed in an integrated research project under
the major research plan of "Integrated research on the ecohydrological process of the
Heihe River Basin" (Yang et al., 2015; Gao et al., 2016). The GBEHM used 1-km grid
system to discretize the study catchment. Based on the 1-km digital elevation model
(DEM), the study catchment was divided into 251 sub-catchments. A sub-catchment
was further divided into flow-intervals along the main stream of the sub-catchment
(Yang et al., 2015). To capture the sub-grid topography, the grid was represented by a
number of hillslopes with an average length and gradient, but different aspect, which
were estimated from the 90-m DEM. The terrain properties of a hillslope include the
slope length and gradient, slope aspect, the soil type and vegetation type (Yang et al.,

2015).





The hillslope is the basic unit for the hydrological simulation, on which the water
and heat transfer (both of conduction and convection) in the vegetation canopy, snow
cover/glacier, soil layers are simulated. The canopy interception, radiation transfer in
the canopy and the energy balance of the land surface are described using the methods
developed in SIB2 (Sellers et al., 1985, 1996). The surface runoff on the hillslope is
solved using the kinematic wave equation. The groundwater aquifer is considered an
individual storage corresponding to each grid. Exchange between the groundwater and
the river water is calculated using Darcy's law (Yang et al., 1998, 2002).
The model runs with a time step of 1 hour. Runoff generated from the grid is the
lateral inflow into the river at the same flow interval in the corresponding sub-
catchment. Flow routing in the river network is calculated using the kinematic wave
equation following the sequence determined by the Horton-Strahler scheme (Yang et
al., 1998, 2015).
**3.2 Simulation of cryospherical processes**
The simulation of cryosphere processes in GBEHM includes glacier ablation, snow
melt, and soil freezing and thawing.
(1) Glacier ablation
Glacier ablation is simulated using an energy balance model (Oerlemans, 2001) as:
$$Q_M = SW(1-\alpha) + LW_{in} - LW_{out} - Q_H - Q_L - Q_G + Q_R \qquad (1)$$
where $Q_M$ is the net energy absorbed by the surface of the glacier (W/m$^2$); $SW$ is the
incoming shortwave radiation (W/m$^2$); $\alpha$ is the surface albedo; $LW_{in}$ is the incoming
longwave radiation (W/m$^2$); $LW_{out}$ is the outgoing longwave radiation (W/m$^2$); $Q_H$ is





the sensible heat flux (W/m$^2$); $Q_L$ is the latent heat flux (W/m$^2$); $Q_R$ is the energy from
rainfall (W/m$^2$); and $Q_G$ is the penetrating shortwave radiation (W/m$^2$). The surface
albedo is calculated as (Oerlemans and Knap, 1998):

$$\alpha = \alpha_{snow} + (\alpha_{ice} - \alpha_{snow})e^{-h/d^*}$$

(2)

where $\alpha_{snow}$ is the albedo of snow on the glacier surface; $\alpha_{ice}$ is the albedo of the ice
surface; $h$ is the snow depth on the glacier surface (m); $d^*$ is a parameter of the snow
depth effect on the albedo (m).

The amount of melt water is calculated as (Oerlemans, 2001):

$$M = \frac{Q_M}{L_f} dt$$

(3)

where $dt$ is the time step used in the model (s) and $L_f$ is the latent heat of fusion (J/kg).
(2) Snow melt

A multi-layer snow cover model is used to describe the mass and energy balance of

snow cover. For each snow layer, temperature is solved using an energy balance
approach (Bartelt and Lehnin, 2002):

$$C_s \frac{\partial T_s}{\partial t} - L_f \frac{\partial \rho_i \theta_i}{\partial t} = \frac{\partial}{\partial z}(K_s \frac{\partial T}{\partial z}) + \frac{\partial I_R}{\partial z} + Q_R$$

(4)

where $C_s$ is the heat capacity of snow (J m$^{-3}$ K$^{-1}$); $T_s$ is the temperature of the snow
layer (K); $\rho_i$ is the density of the ice (kg/m$^3$); $\theta_i$ is the volumetric ice content;
$K_s$ is the thermal conductivity of snow (W m$^{-1}$ K$^{-1}$); $L_f$ is the latent heat of ice fusion
(J/kg) ; $I_R$ is the radiation transferred into the snow layer (W/m$^2$) and $Q_R$ is the energy
brought by rainfall (W/m$^2$) which is only considered for the top snow layer. The solar
radiation transfer in the snow layers and the snow albedo are simulated using the
SNICAR model which is solved using the method developed by Toon et al. (1989). Eq.





(4) is solved using a finite differential scheme.
The mass balance of the snow layer is described as (Bartelt and Lehnin, 2002):

$$\frac{\partial \rho_i \theta_i}{\partial t} + M_{iv} + M_{il} = 0 \tag{5}$$

$$\frac{\partial \rho_l \theta_l}{\partial t} + \frac{\partial U_l}{\partial z} + M_{lv} - M_{il} = 0 \tag{6}$$

where $\rho_l$ is the density of the liquid water (kg/m$^3$); $\theta_l$ is the volumetric liquid water
content; $U_l$ is the liquid water flux (kg m$^{-2}$ s$^{-1}$); $M_{iv}$ is the mass of ice that is changed
into vapor within a time step (kg m$^{-3}$ s$^{-1}$); $M_{il}$ is the mass of ice that is changed into
liquid water within a time step (kg m$^{-3}$ s$^{-1}$); and $M_{lv}$ is the mass of liquid water that is
changed into vapor within a time step (kg m$^{-3}$ s$^{-1}$). The liquid water flux of the snow
layer is calculated as (Jordan, 1991):

$$U_l = -\frac{K_l}{\mu_l} \rho_l^2 g \tag{7}$$

where $K_l$ is the hydraulic permeability (m$^2$), $\mu_l$ is dynamic viscosity of water at 0 ℃
(1.787 × 10$^{-3}$ N s/m$^2$), $\rho_l$ is the density of liquid water (kg/m$^3$) and $g$ is gravitational
acceleration (m/s$^2$). The water flux of the bottom snow layer is considered snowmelt
runoff.
(3) Soil freezing and thawing
The energy balance of the soil layer is solved as (Flerchinger and Saxton, 1989):

$$C_s \frac{\partial T}{\partial t} - \rho_i L_f \frac{\partial \theta_i}{\partial t} - \frac{\partial}{\partial z}(\lambda_s \frac{\partial T}{\partial z}) + \rho_l c_l \frac{\partial q_l T}{\partial z} = 0 \tag{8}$$

where $C_s$ is the volumetric soil heat capacity (J m$^{-3}$ K$^{-1}$); $T$ is the temperature (K) of
the soil layers, $z$ is the vertical depth of the soil (m); $\theta_i$ is the volume ice content; $\rho_i$
is the density the ice (kg/m$^3$); $\lambda_s$ is the thermal conductivity (W m$^{-1}$ K$^{-1}$); $\rho_l$ is the
density of liquid water (kg/m$^3$); and $c_l$ is the heat capacity of liquid water (J kg$^{-1}$ K$^{-1}$).



In addition, $q_l$ is the water flux between different soil layers (m/s) and is solved using
the 1-D Richards equation. The unsaturated soil hydraulic conductivity is calculated
using the modified van Genuchten's equation (Wang et al., 2010) as:
$$K = f_{ice} K_{sat} (\frac{\theta_l - \theta_r}{\theta_s - \theta_r})^{1/2} [1 - (1 - (\frac{\theta_l - \theta_r}{\theta_s - \theta_r})^{-1/m})^m]^2 \qquad (9)$$
where $K$ is the unsaturated soil hydraulic conductivity (m/s); $K_{sat}$ is the saturated soil
hydraulic conductivity; $\theta_l$ is the volumetric liquid water content; $\theta_s$ is the saturated
water content; $\theta_r$ is the residual water content; $m$ is an empirical parameter in van
Genuchten's equation and $f_{ice}$ is an empirical hydraulic conductivity reduction factor
which is calculated using soil temperature as (Wang et al., 2010):
$$f_{ice} = \exp[-10(T_f - T_{soil})], \quad 0.05 \le fice \le 1 \qquad (10)$$
where $T_f$ is 273.15 K and $T_{soil}$ is the soil temperature.
Eq. (8) solves the soil temperature with the upper boundary condition as the heat flux
into the top surface soil layer. When the ground is not covered by snow, the heat flux
from the atmosphere into the top soil layer is expressed as (Oleson et al., 2010):
$$h = S_g + L_g - H_g - \lambda E_g + Q_R \qquad (11)$$
where $h$ is the upper boundary heat flux into the soil layer (W m$^{-2}$); $S_g$ is the solar
radiation absorbed by the top soil layer (W m$^{-2}$); $L_g$ is the net long wave radiation
absorbed by the ground (W m$^{-2}$), $H_g$ is the sensible heat flux from the ground (W m$^{-2}$);
$\lambda E_g$ is the latent heat flux from the ground (W m$^{-2}$); and $Q_R$ is the energy brought by
rainfall (W/m$^2$). When the ground is covered by snow, the heat flux into the top soil
layer is calculated as:
$$h = I_p + G \qquad (12)$$





where $I_p$ is the radiation that penetrates the snow cover, and $G$ is the heat conduction
from the bottom snow layer to the top soil layer. Eq (8) is solved using a finite
differential scheme with an hourly time step.

To simulate the permafrost we consider an underground depth of 50 m and assume

the bottom boundary condition as zero heat flux exchange. The vertical soil column is
divided into 39 layers in the model. The topsoil of 1.7 m is subdivided into 9 layers.
The first layer is 5 cm and the soil layer thickness increases linearly from 5 cm to 30
cm up to the depths of 0.8 m and then decreases linearly to 10 cm up to the depths of
1.7 m. There are 12 soil layers from 1.7 m to 3.0 m with a constant thickness of 10 cm.
From the depth of 3 m to 50 m, there are 18 layers with thickness increasing
exponentially from 10 cm to 12 m. The liquid soil moisture, ice content, and soil
temperature of each layers are calculated at each time step. The soil heat capacity and
soil thermal conductivity are estimated using the method developed by Farouki (1981).
**3.3 Model calibration**

In this study, model simulation during the period of 1961-2001 was used to spin up

to specify the initial values of the hydrological variables (e.g., soil moisture, soil
temperature, soil ice content, groundwater table, etc.). The period of 2002-2006 was
used for model calibration and the period of 2008-2012 was for model validation. The
daily soil temperature at the Qilian station and the frozen depths at the Qilian and
Yeniugou stations were used to calibrate the soil reflectance according to vegetation
type. The other parameters such as groundwater conductivity were calibrated according
to the streamflow discharge in the winter season. We calibrated the surface retention





capacity and surface roughness to match the observed flood peaks, and calibrated the
leaf reflectance, leaf transmittance and maximum Rubsico capacity of the top leaf based
on the remote sensing evapotranspiration data. Table 1 shows the major parameters used
in the model.
**4.  Results**
**4.1 Validation result**
Figure 2 shows the comparison of the model-simulated and observed soil
temperature profiles at six boreholes. The model successfully captured the vertical
distribution of the soil temperature at T1, T2, T3 and T4 in the permafrost area, but
there were some overestimations above 20 m. The errors in simulating the vertical
temperature profile near the surface might be due to simplification of the 3-D
topography. At T5 located in seasonally frozen ground, the simulated soil temperature
profile from approximately 4 m to 20 m does not agree well with the observed one. This
error might be related to the heterogeneity of soil properties especially the thermal
conductivity and heat capacity, which might not be accurately described by the current
data. The model simulation agrees well with the borehole observation at T7, which is
located at the transition zone from permafrost to seasonally frozen ground. This implies
that the model well identified the lower limit of permafrost.
We also validated model simulation of the freezing/thawing cycles based on long-
term observations of soil temperature and frozen depth. Figure 3 compares the
simulated soil temperature with the observed temperature at the Qilian station, which
is located in the seasonally frozen ground (observed daily soil temperature data are





available since 2004). Generally, the model simulations accurately captured the changes
in soil temperature profile. Validation of the soil temperature at different depths (5 cm,
10 cm, 20 cm, 40 cm, 80 cm, 160 cm, and 320 cm) showed that the root mean square
errors decreases with increasing depth. The errors was approximately 3℃ for the top
three depths (5 cm, 10 cm and 20 cm). The error for depths of 40 cm and 80 cm was
2.5℃ and 1.9℃, respectively, and the error was 0.9℃ at a depth of 3.2 m. We
compared the model-simulated daily frozen depth with in-situ observations at the Qilian
and Yeniugou Stations from 2002 to 2014, as shown in Figure 4. The model accurately
reproduced the daily variations in frozen depth although the depth was underestimated
by approximately 50 cm at the Yeniugou station. In general, the validation of soil
temperature and frozen depth indicates that the model well captured the freezing and
thawing processes in the upper Heihe basin.

The observed hourly liquid soil moisture at the A'rou Sunny Slope station was used

as an independent additional validation. Figure 5 shows the comparison between the
simulated and observed liquid soil moisture at different depths from January 1 to
December 31 in 2014. The model simulation agreed well with the observed liquid soil
moisture during the freezing and thawing processes at different depths. However,
relatively larger errors existed in the simulations at a depth of 4 cm, which might be
related to the heterogeneity along the soil column that was not fully addressed in the
model.

Figure 6 compares the model simulated and the observed daily streamflow discharge

at the Yingluoxia, Qilian and Zhamashike station. The model simulation agreed well



with the observations. The model simulation captured the flood peaks and the
magnitude of base flow in both of the calibration and validation periods. In the
calibration period, the Nash-Sutcliffe efficiency (NSE) coefficient was 0.64, 0.65 and
0.70 for the Yingluoxia, Qilian and Zhamashike stations, respectively. In the validation
period, the NSE value were 0.65, 0.60, and 0.75. The relative error (RE) was within 10%
for both the calibration and validation period (see Table 2). Figure 7 shows the
comparison of the model-simulated monthly actual evaporation and remote sensing-
based evaporation data for the entire calibration and validation periods. The GBEHM
simulation showed similar temporal variations in actual evapotranspiration compared
with the remote sensing based estimation, and the root mean square error (RMSE) of
the simulated monthly evapotranspiration was 8.0 mm in the calibration period and 6.3
mm in the validation period. These validation results indicate that the model accurately
simulates the cryosphere hydrological processes in the upper Heihe basin.
**4.2 Long-term changes in freezing-thawing processes and frozen soils**
The freezing-thawing and hydrological processes of the upper Heihe basin from 1961
to 2013 were simulated by GBEHM. A 50-year run which repeated the atmosphere
forcing in the period of 1961-1970 was used to obtain the initial conditions. The long-
term changes in frozen soils, runoff and soil moisture were analyzed based on the model
simulation.
In the upper Heihe basin, the ground surface starts freezing in November and thawing
in April (Wang et al., 2015a). From November to March, the ground surface
temperature is below 0℃ in both the permafrost and seasonally frozen ground regions,



and precipitation mainly falls in the period from April to October. Therefore, a year is
subdivided into two seasons, i.e., the freezing season (November to March) and the
thawing season (April to October) to investigate the changes in frozen soils and their
hydrological impact. Increasing of precipitation and air temperature in the study area in
both seasons in the past 50 years was reported in a previous study (Wang et al., 2015b).

Figure 8 shows the changes in the basin-averaged soil temperature in the freezing

and thawing seasons. The soil temperature increased in all seasons especially in the past
30 years. The increasing trend of soil temperature was larger in the freezing season than
in the thawing seasons. In the freezing season (Figure 8(a)), the top layer soil
temperature was lower than the deep layer soil temperature. The linear trend of the top
layer (0-0.5 m) soil temperature was 0.31℃/10yr and the trend of the deep layer (2.5-3
m) soil temperature was 0.22℃/10yr. The soil temperature in deep layer (2.5-3 m)
changed from -1.1℃ in the 1960s to near 0℃ in the most recent decade. In the thawing
season (see Figure 8(b)), the increasing trend of the top layer (0-0.5 m) soil temperature
(0.17℃/10yr) was greater than the trend of the deep layer (2.5-3 m) soil temperature
(0.10℃/10yr).

Permafrost is defined as ground with a temperature at or below 0℃ for at least two

consecutive years (Woo, 2012). This study differentiated permafrost from seasonally
frozen areas based on the simulated vertical soil temperature profile. For each year, the
frozen soil condition was determined by searching the soil temperature profile within a
four-year window from the previous three years to the current year. Figure 9 shows the
area change of the permafrost during 1961-2013. As shown in Figure 9 (a), the



permafrost areas decreased approximately 9.5% (6445 km$^2$ in the 1970s and 5831 km$^2$
in the 2000s), indicating evident degradation of the permafrost in the upper Heihe basin
in the past 50 years.

Figure 9 (b) shows the changes in the basin-averaged maximum frozen depth for the

seasonally frozen ground and active layer thickness over the permafrost. The basin-
averaged annual maximum frozen depth showed a significant decreasing trend (4.1
cm/10yr). In addition, the maximum frozen depth had a significantly negative
correlation with the annual mean air temperature ($r$ = -0.73). In contrast, an increasing
trend of active layer thickness in the permafrost regions was observed (2.2 cm/10yr),
which had a significantly positive correlation with the annual mean air temperature.

Figure 10 shows the frozen soils distributions in the period of 1971-1980 and in the

period of 2001-2010. Comparing the frozen soils distributions in the two periods, major
changes in frozen soils were observed on the sunny slopes at elevation between 3500
and 3700 m, especially in the west tributary, where large areas of permafrost changed
into seasonally frozen ground.

Figure 11 shows the monthly mean soil temperature over the areas with elevation

between 3300 and 3500 m and over areas with elevation between 3500 and 3700 m in
the upper Heihe basin. In the areas with elevation between 3300 and 3500 m located in
the seasonally frozen ground region, as shown in Figure 11(a), the frozen depth
decreased and the soil temperature in the deep layer (with depth greater than 2 m)
increased. Figure 11(b) shows that the increase in soil temperature was larger in the
area with higher elevation (3500-3700 m). This figure shows that the thickness of the





permafrost layer decreased as soil temperature increased, and the permafrost changed
into seasonally frozen ground after 2000.
**4.3 Changes in the water balance and the hydrological processes**
Table 3 shows the decadal changes in the annual water balance from 1961 to 2010
based on the model simulation. The annual precipitation, annual runoff and annual
runoff ratio had the same decadal variation; however the annual evapotranspiration
maintained an increasing trend since the 1970s which was consistent with the rising air
temperature and soil warming. Although the actual evapotranspiration increased, the
runoff ratio remained stable during the 5 decades because of the increased precipitation.
The changes in runoff (both simulated and observed) in different seasons are shown
in Figure 12 and Table 4. The model-simulated and observed runoff both showed a
significant increasing trend in the freezing season and in the thawing season. This
indicates that the model simulation accurately reproduced the observed long-term
changes. In the freezing season, since there was no glacier melt and snow melt (see
Table 4), runoff was mainly the subsurface flow. In the thawing season, as shown in
Table 4, snowmelt runoff contributed approximately 16% of the total runoff and glacier
runoff contributed only a small fraction of total runoff (approximately 2.4%). Therefore,
rainfall runoff was the major component of total runoff in the thawing season, and the
runoff increasing in the thawing season was mainly due to increased rainfall runoff. As
shown in Figure 12, the actual evapotranspiration increased significantly in both
seasons due to increased precipitation and soil warming. The increasing trend of the
actual evapotranspiration was higher in the thawing season than in the freezing season,



which indicates that the actual evapotranspiration was limited by the water available in
this region.
Figure 13 shows the changes in the basin-averaged annual water storage in the top
0-3 m layer and the groundwater storage. The annual liquid water storage of the top 0-
3 m showed a significant increasing trend especially in the most recent 3 decades. This
long-term change in liquid water storage was similar to the runoff change in the freezing
season, as shown in Figure 12 (a), with a correlation coefficient of 0.80. The annual ice
water storage in the top 0-3 m soil showed significant decreasing trend due to frozen
soils changes. Annual groundwater storage showed a significantly increasing trend
especially in the most recent 3 decades, which indicates the groundwater recharge
increases with the frozen soil degradation.
**5.  Discussion**
**5.1 Impact of frozen soils changes on the soil moisture and runoff**
Figure 14 shows the spatial-averaged liquid soil moisture changes in the region
covered by seasonally frozen ground with elevation between 3300 and 3500 m and in
the area with elevation between 3500 and 3700 m where the permafrost changed into
seasonally frozen ground. In the seasonally frozen ground with elevation of 3300-3500
m (Figure 14(a)), by comparing with the soil temperature shown in Figure 11 (a), we
can see that the liquid soil moisture increase was mainly caused by the decrease in the
frozen depth. The liquid soil moisture in the deep soil layer increased significantly since
1990s (see Figure 14(b)) in the area with elevation of 3500-3700 m where the
permafrost changed to seasonally frozen ground. Compared with the soil temperature



change shown in Figure 11 (b), the liquid soil moisture increases in this region was
mainly caused by the change of permafrost to seasonally frozen ground, indicating that
the frozen soils degradation caused a significant increase in liquid soil moisture.
Therefore, the basin-averaged liquid soil moisture was highly correlated with the soil
temperature in the freezing seasons as shown in Table 5. The liquid soil moisture was
also highly correlated with soil temperature in the thawing season, because of the
increase in the active layer thickness of the permafrost and degradation of the
permafrost (i.e., the change from permafrost to seasonally frozen ground). This
correlation was larger than the correlation between liquid soil moisture and
precipitation because the liquid soil moisture increase caused by the permafrost
degradation is more significant than the liquid soil moisture increase caused by
increased precipitation in the thawing season.
In the freezing season, since the surface ground is frozen, runoff is mainly subsurface
flow coming from seasonally frozen ground. Table 5 shows that runoff has the highest
correlation with the liquid soil moisture in the freezing season, which indicates that the
frozen soils change was the major cause of the increased liquid soil moisture, resulting
in increased runoff in the freezing season. During the past 50 years, parts of the
permafrost changed into seasonally frozen ground, and the thickness of the seasonally
frozen ground decreased, which led to increased liquid soil moisture in the deep layers
during the freezing season as shown in Figure 14. The increase in liquid soil moisture
also increased the hydraulic conductivity which enhanced the subsurface flow.
In the thawing season from April to October, the thickness of the seasonally frozen





ground rapidly decreased to zero and the thaw depth of permafrost reached the
maximum. Runoff in the thawing season was mainly rainfall runoff as shown in Table
4. Table 5 shows that runoff was more strongly correlated with precipitation and
relatively more weakly correlated with liquid soil moisture, which illustrates that the
increased runoff mainly came from increased precipitation in the thawing season. The
correlation between runoff and liquid soil moisture in the thawing season was mainly
due to the high correlation between the liquid soil moisture and the precipitation.

Figure 15 shows the changes in areal mean runoff along the elevation for different

seasons. There was a large difference in runoff variation with the elevation during the
different seasons. In the freezing season, the runoff change from the 1970s to the 2000s
in the region of seasonally frozen ground (mainly located below 3500 m, see Figure 10)
was relatively small. Runoff in the areas with elevation of 3500-3900 m showed larger
change. This is due to the shift from permafrost to seasonally frozen ground in some
areas with elevation range of 3500-3900 m as simulated by the model, particularly for
the sunny hillslopes (see Figure 10). This illustrates that a change from the permafrost
to the seasonally frozen ground has a larger impact on the runoff than a change in frozen
depth in seasonally frozen ground. In the thawing season runoff increased with
elevation due to the increase in precipitation with increasing elevation, and the runoff
increase was mainly determined by increased precipitation (Gao et al., 2016).
Precipitation in the region with elevation below 3100 m was low but air temperature
was high. Runoff in this region decreased during 2001-2010 compared to 1971-1980
because of higher evapotranspiration.



### 5.2 Comparison with the previous similar studies

In this study, the model simulation showed that changes in frozen soils led to increased freezing season runoff and base flow in the upper Heihe basin. This result is consistent with previous findings based on the trend analysis of streamflow observations in high latitude regions (Walvoord et al., 2016; Jacques and Sauchyn, 2009; Ye et al., 2009) and in Northeast China (Liu et al., 2003). However, those studies lacked of spatial variability. This study found that the impact of the change in frozen soils on runoff had regional characteristics. In the upper Heihe basin (see Figure 15), a change in frozen soils led to the increased runoff at higher elevations but led to decreased runoff at lower elevation region during the freezing season. This implies that change of the freezing season runoff was controlled by the permafrost degradation in higher elevation region but by the evaporation increase in the lower elevation region due to the air temperature rising. However, runoff at the basin scale mainly came from the higher elevation regions.

This study also showed that the change in frozen soils increased the soil moisture in the upper Heihe basin, which is consistent with the finding of Subin et al. (2013) using the CLM model simulation in north latitude permafrost regions, and the findings of Cuo et al. (2015) using VIC model simulation at 13 sites on the Tibetan Plateau. However, Lawrence et al. (2015) found that permafrost thawing caused soil moisture drying based on CLM model simulations for the global permafrost region. This might be related to the uncertainties in the soil water parameters and the highly spatial heterogeneity of soil properties, which are difficult to consider in a global-scale model. Subin et al. (2013)





and Lawrence et al. (2015) modelled the soil moisture changes in the active layer of
permafrost in large areas with coarse spatial resolution. This study revealed the spatio-
temporal variability of soil moisture with high spatial resolution and analyzed the
correlations with the change in frozen soils.
Wu and Zhang (2010) focused on the changes in the active layer thickness at 10 sites
in the permafrost region on the Tibetan Plateau and found a significant increasing trend
during the period of 1995-2007, which is consistent with the result of this study. Jin et
al. (2009) found decreased soil moisture and runoff due to the permafrost degradation
based on observation at the plot scale in the source areas in the Yellow River basin. This
result is different from the present study, possibly due to the difference of
hydrogeological structure and the soil hydraulic parameters in the source area of Yellow
River from those in the upper Heihe basin. Wang et al. (2015a) focused on the change
in the seasonally frozen ground in the Heihe River basin based on plot observations,
and the increasing trend of the maximum frozen depth was estimated as 4.0 cm/10yr
during 1972-2006, which is consistent with the GBEHM model simulation in this study.
The increase in groundwater storage illustrated in this study is also consistent with the
finding of Cao et al. (2012) based on the GRACE data which showed that groundwater
storage increased during the period of 2003~2008 in the upper Heihe basin.
**5.3 Uncertainty in the frozen soil simulation**
Estimation of the change in permafrost area is a great challenge due to the complex
climatology, vegetation, geology. Different methods produce large differences in their
estimation results. Jorgenson et al. (2006) found a 4.4% decrease in the area of





permafrost in Arctic Alaska from 1982 to 2001 based on airphotos analysis. Wu et al.
(2005) reported that the permafrost area decreased by 12% from 1975 to 2002 in the
Xidatan basin, Qinghai-Tibetan Plateau based on a ground penetration radar survey. Jin
et al. (2006) found an area reduction of 35.6% in island permafrost in Liangdaohe,
which is located at the southern Qinghai–Tibet Highway, from 1975 to 1996. Chasmer
et al. (2010) found a 30% reduction of the discontinuous permafrost area in the
Northwest Territories, Canada from 1947 to 2008 based on remote sensing. This study
conducted an integrated simulation of permafrost change and regional hydrological
change. Compared with the site observation of Wang et al. (2013) shown in Figure 2,
this model slightly overestimated the soil temperature in permafrost areas, which might
lead to overestimation of the rate of permafrost area reduction.
There were two major uncertainties in the frozen soils simulation: uncertainty in the
land surface energy balance simulation and uncertainty in the simulation of the soil
heat-water transfer processes. Uncertainty in the land surface energy balance simulation
might result from the estimations of radiation and surface albedo due to the complex
topography, vegetation cover and soil moisture distribution, which may induce
uncertainties in the estimated ground temperature and thermal heat flux into the deep
layers. The uncertainty in simulation of soil heat-water transfer processes might result
from the soil water and heat parameters and the bottom boundary condition of heat flux.
Permafrost degradation is closely related to the thermal properties of rocks and soils,
geothermal flow and initial soil temperature and soil ice conditions. The lack of
observed initial condition data could also cause uncertainty in the permafrost change



estimation.
**6.  Conclusion**
A distributed hydrological model coupled with cryospherical processes was developed
in the upper Heihe basin. The model was validated using available observations of soil
moisture, soil temperature, frozen depth, and streamflow discharge and was compared
with remote sensing based estimation of actual evapotranspiration. Based on the model
simulation from 1961 to 2013, the changes in frozen soils and the effect of the frozen
soils change on hydrological processes were examined. The conclusions derived in this
study are:
(1) The distributed hydrological model developed in this study accurately simulated
the cryosphere hydrological processes in the upper Heihe basin, and can be used to
analyze change in frozen soils and the impacts on hydrological processes on the high
and cold plateau.
(2) Significant degradation of frozen soils was found in the upper Heihe basin due to
the increasing air temperature over the last 50 years. The permafrost area decreased by
9.5% in the period of 1961-2013 and changed into seasonally frozen ground, especially
in areas at elevation between 3500 m and 3900 m. The annual maximum frozen depth
showed a significant decreasing trend of 4.1 cm/10yr in the seasonally frozen ground,
and the active layer thickness increased 2.2 cm/10yr in the permafrost regions.
(3) In the freezing season (November-March), runoff was mainly subsurface flow
which increased significantly in the higher elevation region due to the change in frozen
soils during the study period. In the thawing season (April-October), runoff mainly

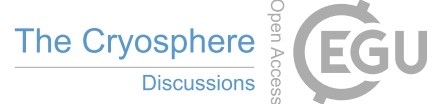

came from rainfall and showed an increasing trend at the higher elevations due to the
increased precipitation. In both the freezing and thawing seasons, runoff decreased in
the lower elevation region due to increased evaporation caused by rising air temperature.
Since the runoff at the basin scale is mainly from the higher elevation regions, annual
runoff showed a significant increasing trend due to the increased precipitation, and the
base flow increased due to the degradation of frozen soils in the study period.
(4) Annual liquid water storage showed a significant increasing trend especially in
the most recent three decades, due to the change in frozen soils. Annual ice water
storage in the top 0-3 m of soil showed a significant decreasing trend due to soil
warming. Annual groundwater storage had an increasing trend, which indicated that
groundwater recharge was enhanced in the last 50 years.
(5) Regions where the permafrost changed into the seasonally frozen ground showed
larger changes in runoff and soil moisture than area covered by seasonally frozen
ground at low elevations.
There were uncertainties in the frozen soils and the hydrological processes
simulations that might be related to the soil properties, the high spatial heterogeneity,
the parameterization of the lower boundary of deep soils, which was important for
simulating the permafrost thawing process, and the other factors. In addition, the
interactions between the change in frozen soils, vegetation dynamics and hydrological
processes need to be investigated in the future study to better understand the change in
ecohydrological processes.



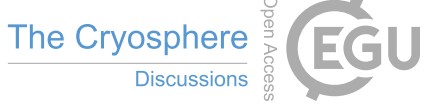

**Acknowledgements:** This research was supported by the major plan of "Integrated
Research on the Ecohydrological Processes of the Heihe Basin" (Project Nos.
91225302 and 91425303) funded by the National Natural Science Foundation of China
(NSFC). The authors would like to thank the editor for their constructive comments,
which greatly improved the manuscript.

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



**Figure caption:**


Figure 1. The Study area, hydrological stations, borehole observation and flux tower stations
Figure 2. Comparison of the simulated and the observed soil temperature at borehole observation
sites, and the observed data is provided by Wang et al. (2013)
Figure 3. Daily soil temperature at the Qilian station: (a) observation; (b) simulation; (c) Simulation-
Observation
Figure 4. Comparison of the simulated and observed daily frozen depths during the period of 2002-
2014 at: (a) the Qilian station, (b) the Yeniugou station
Figure 5. Comparison of the simulated and the observed hourly liquid soil moisture at the A'rou
Sunny Slope station
Figure 6. Comparison of the simulated and the observed daily river discharge at: (a) the Yingluoxia
Gauge, (b) the Qilian Gauge, and (c) the Zhamashike Gauge.
Figure 7. Comparison of the simulated and the remote sensing estimated actual evapotranspiration
in the period of 2002~2012
Figure 8. Changes of the mean soil temperature in different seasons: (a) the freezing season (from
November to March) (b) the thawing season (from April to October)
Figure 9. Change of the frozen soils in the upper Heihe basin: (a) areas of permafrost and basin
averaged annual air temperature; (b) the basin averaged annual maximum frozen depth of the
seasonally frozen ground and the annual maximum thaw depth of the permafrost
Figure 10. Figure 10. Distribution of permafrost and seasonally frozen ground: (a) distribution in
the period of 1971-1980; (b) distribution in the period of 2001-2010; (c) percentage of areas of
permafrost and seasonally frozen ground at sunny slope; (d) percentage of areas of permafrost and





seasonally frozen ground at shaded slope (the same legend as (c))
Figure 11. Spatial averaged monthly soil temperature during the period of 1961-2013 in different
elevation intervals: (a) the seasonally frozen ground with elevation between 3300-3500 m; (b) the
areas where permafrost changed to seasonally frozen ground with elevation between 3500-3700 m
Figure 12. Changes of the runoff and actual evapotranspiration: (a) in the freezing season; (b) in the
thawing season
Figure 13. Changes of the annual water storage (equivalent water depth) during the period of 1961-
2013: (a) the liquid soil water storage of the top 0-3 m layer; (b) the ice water storage of the top 0-
3 m layer; (c) the groundwater storage
Figure 14. Spatial averaged monthly liquid soil moisture during the period of 1961-2013 in different
elevation intervals: (a) the seasonally frozen ground with elevation between 3300-3500 m; (b) the
areas where permafrost changed to seasonally frozen ground with elevation between 3500-3700 m
Figure 15. Model simulated changes of runoff: (a) in the freezing season, (b) in the thawing season





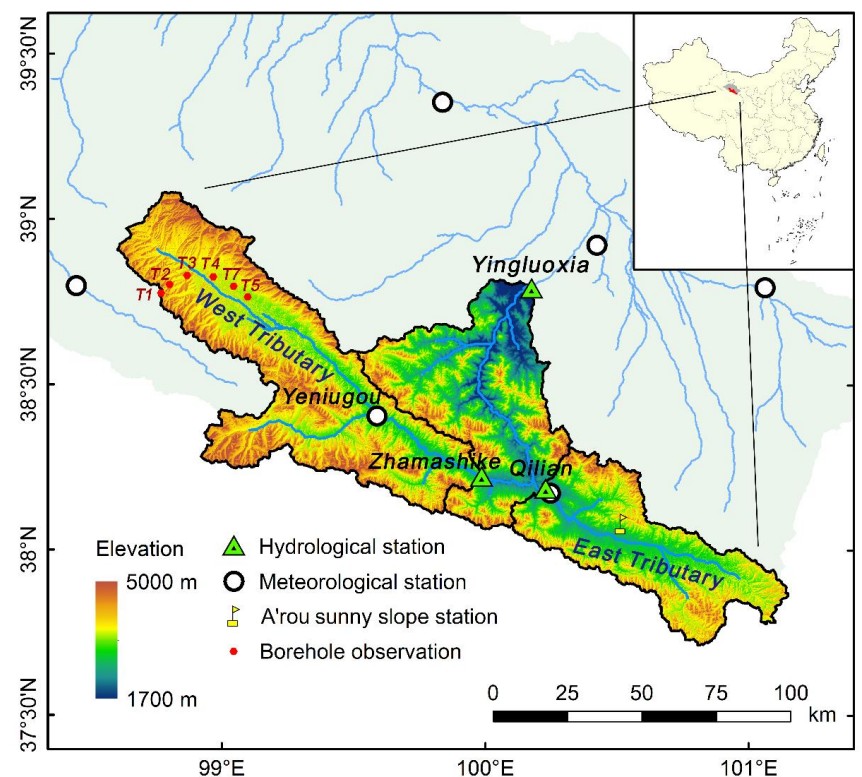

Figure 1. The Study area, hydrological stations, borehole observation and flux tower

stations





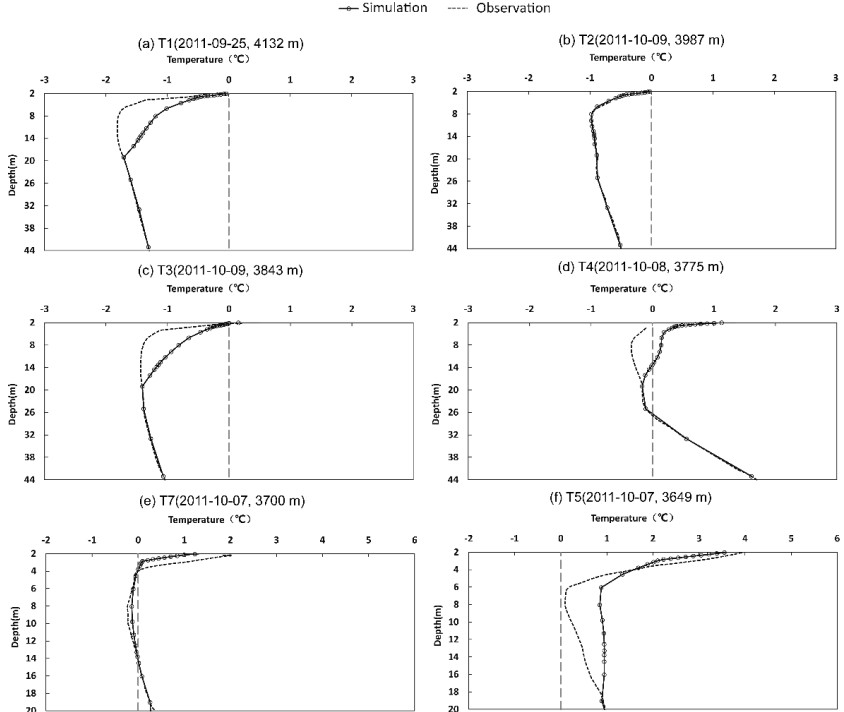

Figure 2. Comparison of the simulated and the observed soil temperature at borehole

observation sites, and the observed data is provided by Wang et al. (2013)




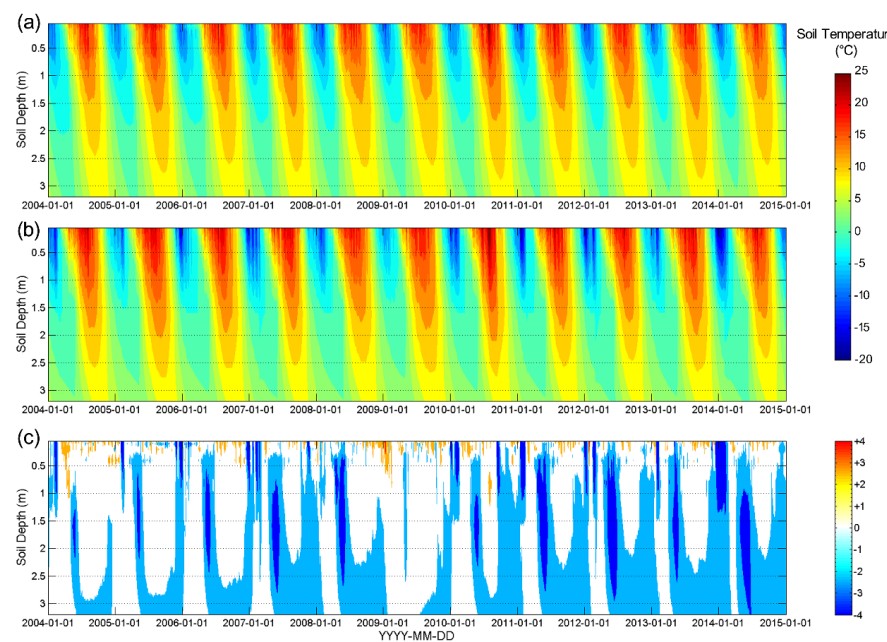

Figure 3 Daily soil temperature at the Qilian station: (a) observation; (b) simulation;

(c) Simulation-Observation



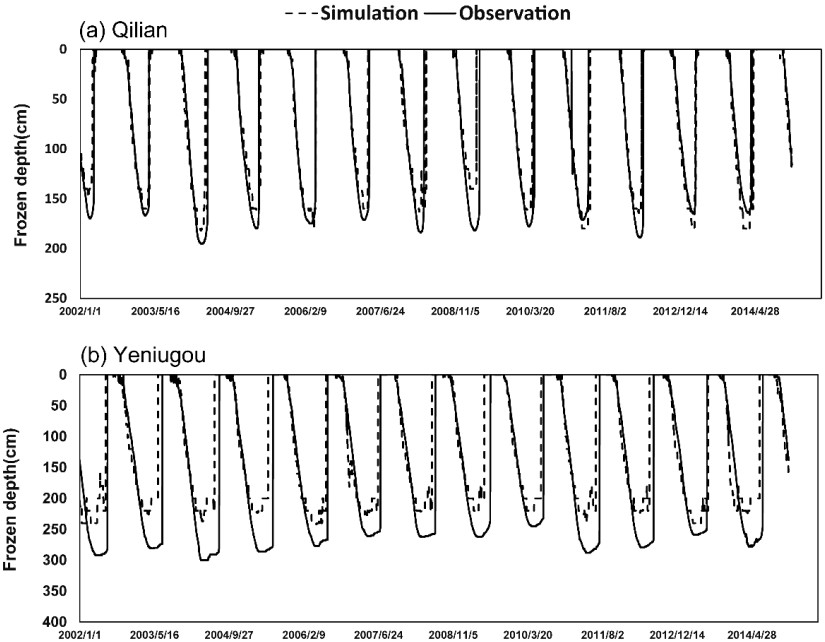


Figure 4. Comparison of the simulated and observed daily frozen depths during the

period of 2002-2014 at: (a) the Qilian station, (b) the Yeniugou station


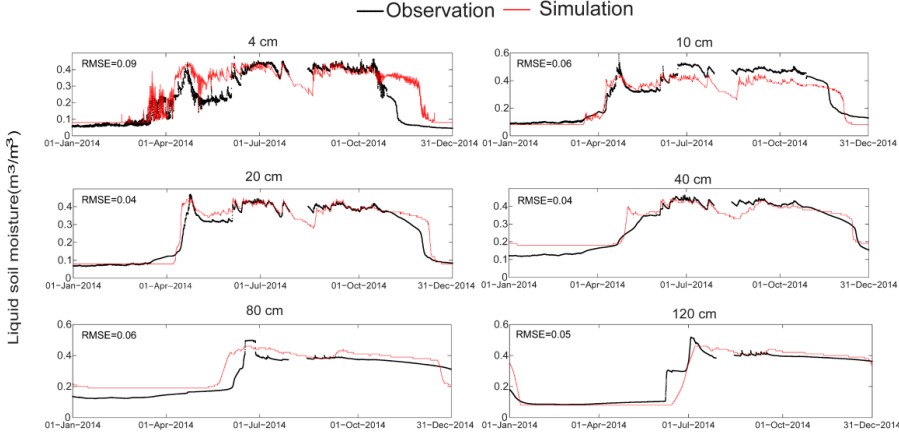


Figure 5. Comparison of the simulated and the observed hourly liquid soil moisture at

the A'rou Sunny Slope station




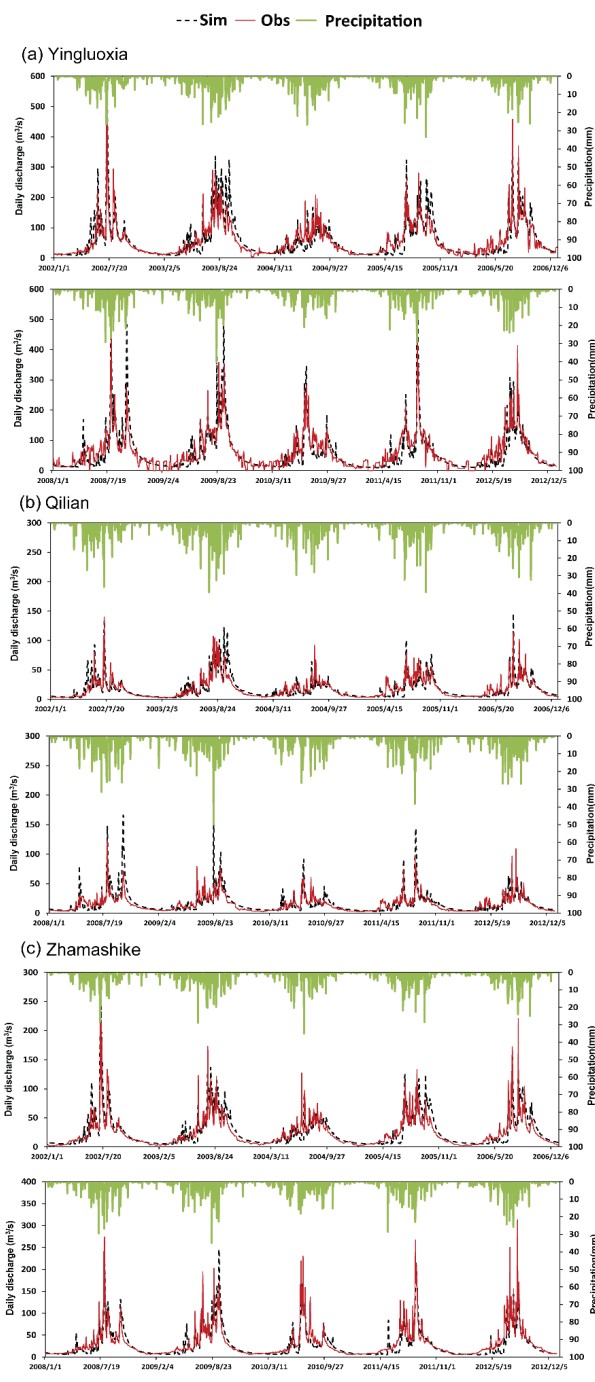


Figure 6. Comparison of the simulated and the observed daily river discharge at: (a)

the Yingluoxia Gauge, (b) the Qilian Gauge, and (c) the Zhamashike Gauge.







Figure 7. Comparison of the simulated and the remote sensing estimated actual

evapotranspiration in the period of 2002~2012








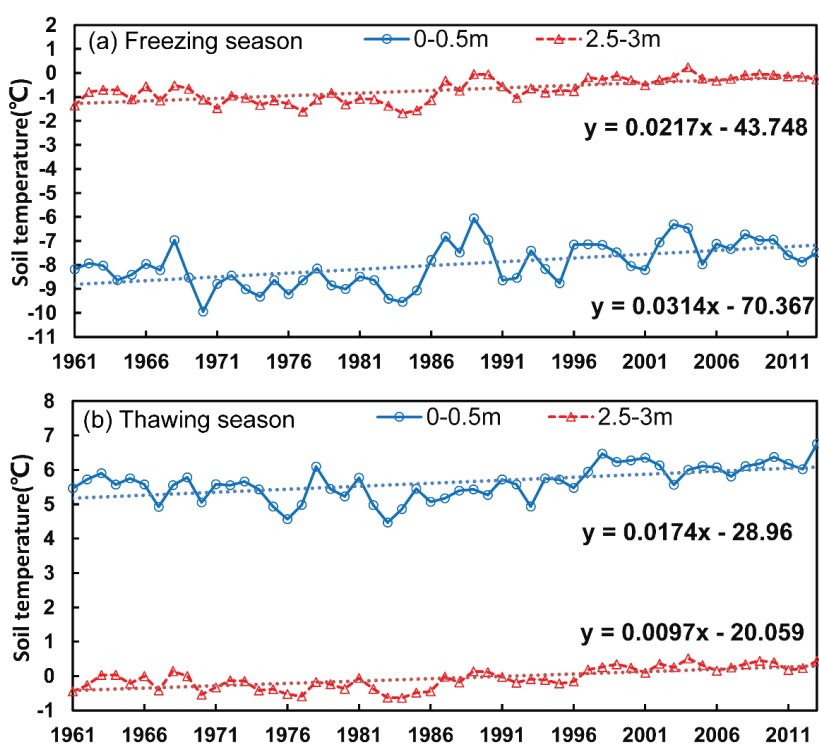


Figure 8. Changes of the mean soil temperature in different seasons: (a) the freezing

season (from November to March) (b) the thawing season (from April to October)






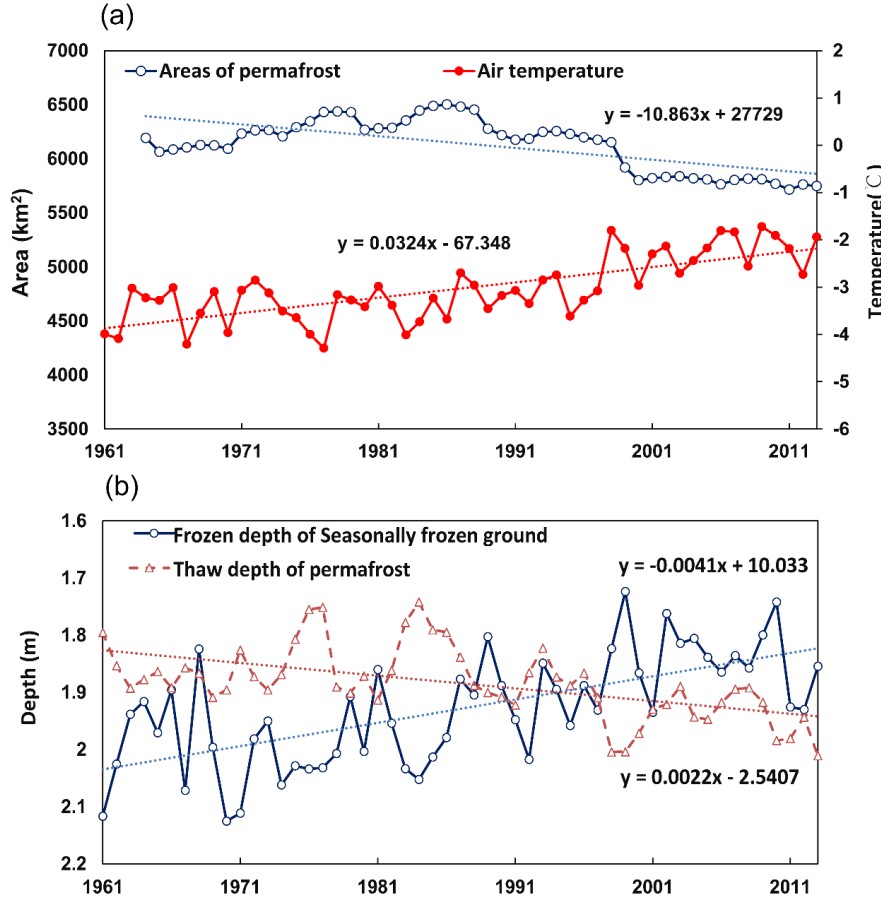


Figure 9. Change of the frozen soils in the upper Heihe basin: (a) areas of permafrost
and basin averaged annual air temperature; (b) the basin averaged annual maximum
frozen depth of the seasonally frozen ground and the annual maximum thaw depth of

the permafrost




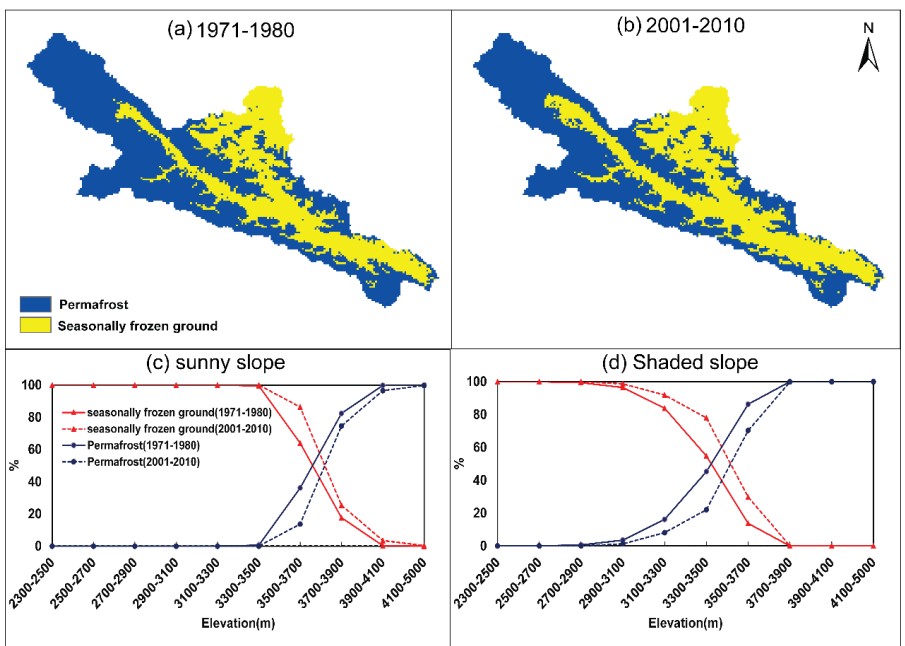

Figure 10. Distribution of permafrost and seasonally frozen ground: (a) distribution in

the period of 1971-1980; (b) distribution in the period of 2001-2010; (c) percentage of

areas of permafrost and seasonally frozen ground at sunny slope; (d) percentage of

areas of permafrost and seasonally frozen ground at shaded slope (the same legend as

(c))



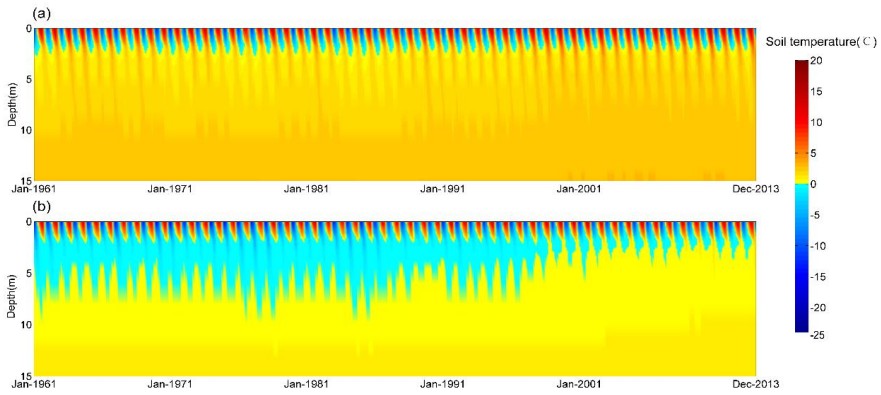

Figure 11. Spatial averaged monthly soil temperature during the period of 1961-2013

in different elevation intervals: (a) the seasonally frozen ground with elevation

between 3300-3500 m; (b) the areas where permafrost changed to seasonally frozen

ground with elevation between 3500-3700 m

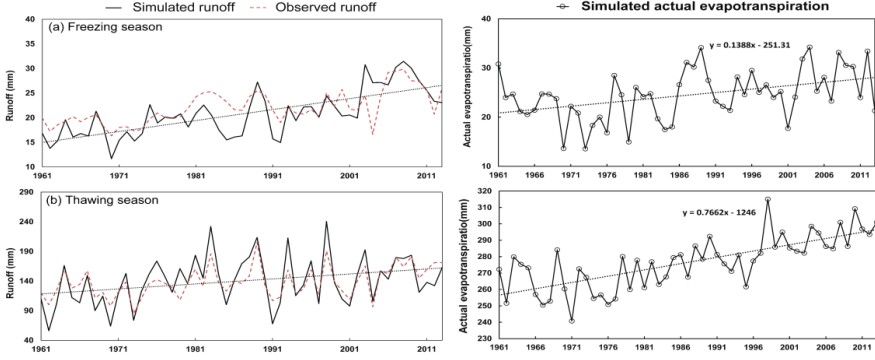

Figure 12. Changes of the runoff and actual evapotranspiration: (a) in the freezing

season; (b) in the thawing season





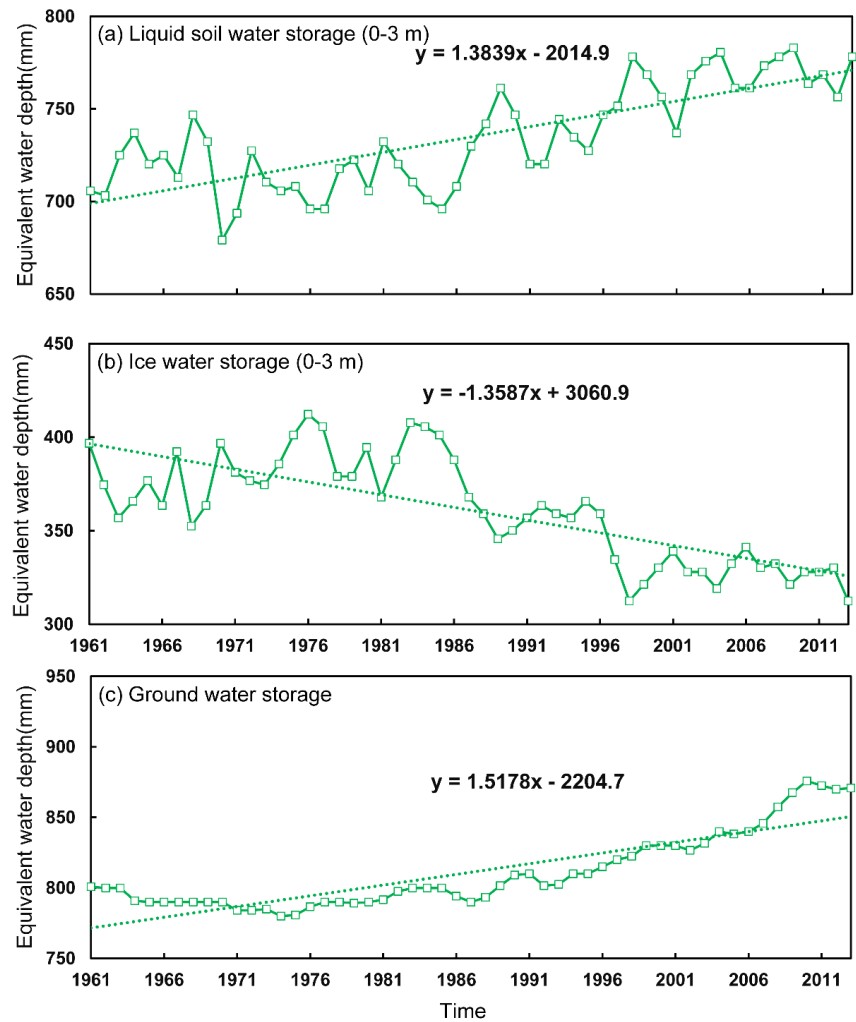


Figure 13. Changes of the annual water storage (equivalent water depth) during the

period of 1961-2013: (a) the liquid soil water storage of the top 0-3 m layer; (b) the ice

water storage of the top 0-3 m layer; (c) the groundwater storage



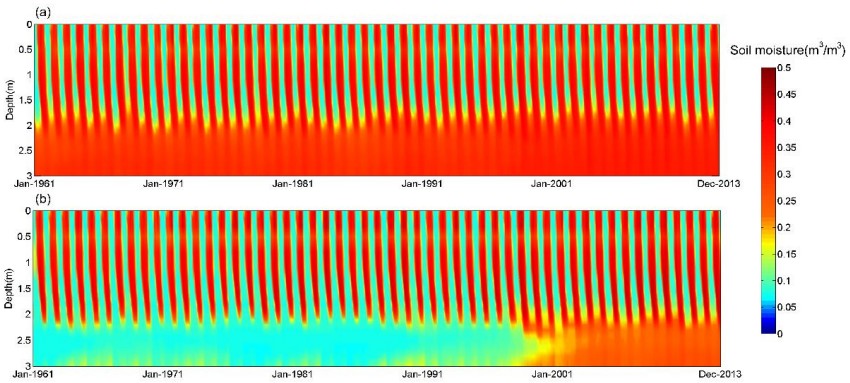


Figure 14. Spatial averaged monthly liquid soil moisture during the period of 1961-
2013 in different elevation intervals: (a) the seasonally frozen ground with elevation
between 3300-3500 m; (b) the areas where permafrost changed to seasonally frozen

ground with elevation between 3500-3700 m




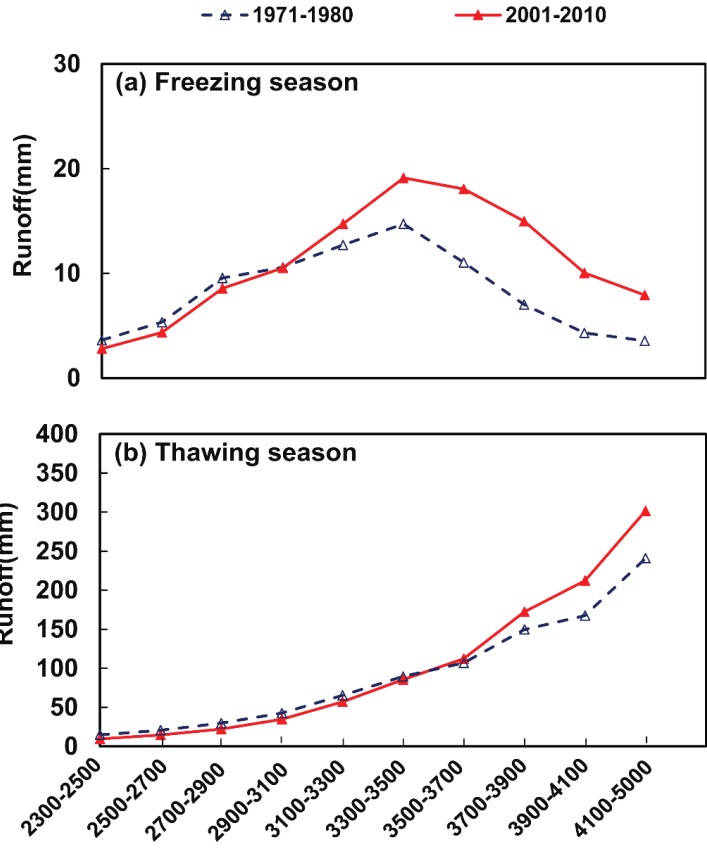

Figure 15. Model simulated changes of runoff: (a) in the freezing season, (b) in the

thawing season





**Table list:**

**Table list:**
Table 1 Major parameters of the GBEHM model
Table 2 Model performance of the daily streamflow simulation
Table 3 Changes in basin water balance
Table 4 Changes in runoff components in different seasons
Table 5 Correlation between runoff/soil moisture and precipitation/soil temperature



Table 1 Major parameters of the GBEHM model

| Parameters | Coniferous Forest | Shrub | Steppe | Alpine Meadow | Alpine Sparse Vegetation | Desert |
|---|---|---|---|---|---|---|
| Surface retention capacity (mm) | 30.0 | 25.0 | 10.0 | 15.0 | 15.0 | 5.0 |
| Surface roughness (Manning coefficient) | 0.5 | 0.3 | 0.1 | 0.1 | 0.1 | 1.0 |
| Soil reflectance to visible light | 0.20 | 0.20 | 0.20 | 0.28 | 0.14 | 0.11 |
| Soil reflectance to near-infrared radiation | 0.225 | 0.225 | 0.225 | 0.28 | 0.225 | 0.225 |
| Leaf reflectance to visible light | 0.105 | 0.105 | 0.105 | 0.105 | 0.105 | — |
| Leaf reflectance to near-infrared radiation | 0.35 | 0.58 | 0.58 | 0.58 | 0.58 | — |
| Leaf transmittance to visible light | 0.05 | 0.07 | 0.07 | 0.07 | 0.07 | — |
| Leaf transmittance to near-infrared radiation | 0.10 | 0.25 | 0.25 | 0.25 | 0.25 | — |
| Maximum Rubsico capacity of top leaf ($10^{-5}$ mol m$^{-2}$ s$^{-1}$) | 6.0 | 6.0 | 3.3 | 3.3 | 3.0 | — |
| Plant root depth (m) | 2.0 | 1.0 | 0.40 | 0.40 | 0.1 | 0.0 |
| Intrinsic quantum efficiency (mol mol$^{-1}$) | 0.08 | 0.08 | 0.05 | 0.05 | 0.05 | — |
| Canopy top height (m) | 9.0 | 1.9 | 0.3 | 0.3 | 0.2 | — |
| Leaf length (m) | 0.055 | 0.055 | 0.3 | 0.3 | 0.04 | — |
| Leaf width (m) | 0.001 | 0.001 | 0.005 | 0.005 | 0.001 | — |
| Stem area index | 0.08 | 0.08 | 0.05 | 0.05 | 0.08 | — |


Table 2 Model performance of the daily streamflow simulation

| Station | Calibration period (2002~2006) | | Validation period (2008~2012) | |
|---|---|---|---|---|
| | NSE | RE (%) | NSE | RE (%) |
| Yingluoxia | 0.64 | 3.8 | 0.65 | -5.6 |
| Qilian | 0.65 | 1.5 | 0.60 | 9.3 |
| Zhamashike | 0.70 | 9.9 | 0.75 | -7.0 |




Table 3 Changes in basin water balance

| Decade | Precipitation (mm/yr) | Actual evaporation (mm/yr) | Simulated runoff (mm/yr) | Observed runoff (mm/yr) | Runoff ratio(observed) | Runoff ratio (simulated) |
|---|---|---|---|---|---|---|
| 1961-1970 | 405.7 | 288.8 | 133.3 | 144.1 | 0.36 | 0.33 |
| 1971-1980 | 439.1 | 280.8 | 154.5 | 143.8 | 0.33 | 0.35 |
| 1981-1990 | 492.8 | 300.0 | 186.2 | 174.1 | 0.35 | 0.38 |
| 1991-2000 | 471.0 | 306.1 | 160.1 | 157.4 | 0.33 | 0.34 |
| 2001-2010 | 504.3 | 317.4 | 177.9 | 174.3 | 0.35 | 0.35 |


Table 4 Changes in runoff components in different seasons

| Freezing season (from November to March) | | |
|---|---|---|
| Total runoff (mm) | Glacier runoff (mm) | Snowmelt runoff (mm) |
| 1961-1970 | 16.5 | 0.0 | 0.0 |
| 1971-1980 | 18.5 | 0.0 | 0.0 |
| 1981-1990 | 20.2 | 0.0 | 0.0 |
| 1991-2000 | 20.4 | 0.0 | 0.0 |
| 2001-2010 | 27.2 | 0.0 | 0.0 |
| Thawing season (from April to October) | | |
| Total runoff (mm) | Glacier runoff (mm) | Snowmelt runoff (mm) |
| 1961-1970 | 116.8 | 3.0 | 26.2 |
| 1971-1980 | 136.0 | 3.5 | 13.5 |
| 1981-1990 | 166.1 | 3.1 | 28.2 |
| 1991-2000 | 139.7 | 3.8 | 19.2 |
| 2001-2010 | 150.7 | 3.7 | 25.8 |






Table 5 Correlation between runoff/soil moisture and precipitation/soil temperature

|  | Freezing season | | | Thawing season | | |
|---|---|---|---|---|---|---|
|  | P | Tsoil | LSM | P | Tsoil | LSM |
| LSM | 0.26 | 0.89 | - | 0.61 | 0.85 | - |
| Runoff | 0.30 | 0.66 | 0.82 | 0.93 | 0.06 | 0.43 |

Note: P is the precipitation, Tsoil is the mean soil temperature of 0-3 m, LSM is the mean liquid soil
moisture of 0-3 m.