# Peer review of "Upper Heihe Basin, the Northeast Qinghai-Tibetan Plateau"

_The Cryosphere, 2016_

## Referee Comment (RC1) · Anonymous Referee #1 · 8 Feb 2017

General comments

Gao et al. present a very impressive study that investigates how changing climate conditions influence hydrologic processes in a cold environment (particularly the Heihe basin in the Qinghai-Tibetan Plateau). Their model enhancements include algorithms for considering frozen ground processes. As such, they address how the hydrologic system responds to forcing (changing precipitation) but also to the changing system characteristics themselves (i.e. type and distribution of frozen ground). This study certainly has merit for publication in TC and has potential to be an important paper in this field. There are a few issues that I think should be considered prior to publication. Most of these are relatively minor, but addressing them all will likely warrant a major

revision.

Major comments

1. I think the introduction should be improved. (1) In the first few paragraphs, the authors go from regional to global and back to regional. I find this confusing. It would be preferable to start global and then narrow down to the Qinghai-Tibetan Plateau. It should be rewritten. (2) There are insufficient references, especially in the second half of the first paragraph. Many of those statements are not self-evident and should be backed up by more studies. I recommend the authors consider the already cited Walvoord and Kurylyk (2016, VZJ) review and references therein to back up these claims about how frozen soil and its evolution influence hydrological processes.

2. There are quite a few places where English issues occur. In general, the manuscript is in pretty good shape though. I do note a number of specific English corrections below in my minor comments.

3. I'm curious if the authors have any citations to provide for the GBEHM model other than the Yang et al. 2015 paper published in a Chinese journal. The description in Section 3.1 is a bit brief, and there is no English source to go to for more information. One question: how are heads calculated in the aquifer since Darcy's Law drives GW-SW exchange?

4. Equation 4. I'm a bit confused on the snowmelt equation, but I'm more used to frozen ground modeling. Why does the latent heat term only allow for ice to water transition and not snow to water? Is this ice referring to firn? This should be more explicit.

5. It is stated that Equations 4 and 8 are solved by the 'finite differential method'. Do the authors mean 'finite difference method'? If so, what method? Are the spatial derivatives solved as forward, backward or centered finite difference? Please state in the text. Is the time derivative implicit or explicit finite difference? Is there a Crank-Nicholson scheme employed? Just one or two more sentences will suffice.

[Figure]

6. The authors would do well to include a figure of their modeling domain – at least their vertical discretization and boundary conditions. I find this lacking.

7. I realize this contradicts the statement above, but this is a bit long of a paper and there are many figures! I went through them and found them mostly valuable, however. I do think that Table 5 could be cut without much of a loss and Figure 14 could be as well. If the authors are in love with these, they could move them to an electronic supplement.

8. I'm confused at how the authors formed permafrost in 50 year spin up run (e.g. L297 and elsewhere). I've modeled even thin permafrost using 1000 year spin ups. Can the authors please describe this better? I wonder if this lack of permafrost formation time might partially cause some of the poor fits seen in Figure 2. Related to this, I wonder if the fits in Figure 2 in the shallow zone would look better at another time in the year (which the authors likely don't have data for). In other words, it could be the seasonal dynamics that are off, not the multi-decadal dynamics.

9. L330 and surrounding text, the authors suggest that RMSE decreases with increasing depth in Figure 3. It looks to me however (somewhat counter to what intuition would suggest) that the errors are increasing with depth (simulation – observation in Figure 3c goes to dark blue at the bottom indicating higher error and is white, indicating low error, higher up).

10. The model results are impressive in Figure 4 (no small feat to get seasonal thaw and frost depths so well captured) and Figure 5 and Figure 6, so I commend the modelers on this.

11. L368. The authors should explain why the warming rate is higher in the shallower soils. It has to do with the surface signal arriving as a 'thermal breakthrough curve' that is retarded at greater depths due to the thermal inertia (sensible and latent heat storage) of the subsurface.

12. I don't understand the point of presenting some of the correlation information. It is tautological to say that frost depth and thaw depth are correlated or inversely correlated to mean annual air temperature (L398-402). Along these lines, Table 5 is also not that useful as already stated. L463-470 is also a waste of space. You don't have to talk about correlation when you are using a physically-based model. Talk about physical processes!

13. L429 and elsewhere, the authors talk about subsurface flow a bit confusingly. Is this groundwater flow? Or is it groundwater flow plus lateral flow in the unsaturated zone? In reality all the precipitation typically makes its way to the river via subsurface flow paths as Hortonian overflow is very rare except in urbanized watersheds with impermeable pavement. They should be a bit clearer if they are talking about groundwater or what. L479 says that higher moisture increases conductivity and thus subsurface flow. True. But again, what flow? If they are talking about groundwater flow, they could have higher lateral transmissivity, but that effect wouldn't be captured in their model, I don't think.

14. L600-601, is it possible that the increased liquid groundwater storage came from phase change of ice to water, rather than increased recharge? Couldn't the authors just directly determine the recharge from their modeling rather than making inferences based on groundwater storage?

Minor comments

L20, insert 'a' before 'regional'

L47, Comma after 'soils'

L52, delete 'balance' after water

L59 and 507, The Walvoord et al. (2016) study (by the way there are no cited Walvoord papers with more than two authors) should really be Walvoord and Striegl 2007 or one of her related studies as these are the original field studies.

[Figure]

e.g. Walvoord and Striegl 2007. Increasing groundwater to stream discharge from permafrost thawing in the Yukon River basin: Potential impacts on lateral export of carbon and nitrogen. Geophys. Res. Lett.

L60, The authors miss a very recent study in NE China that is certainly related to the present study Duan et al., 2017, Increasing winter baseflow in response to permafrost thaw and precipitation regime shifts in northeastern China. Water, 9(1).

L61, The authors suggest a 'few studies' argue this point, but only provide 1 citation. Either provide more citations, use 'e.g.' in the citation, or reword this slightly.

L62, These studies typically included long term hydrological (not just meteorological) data

L63, Change 'might lose' to 'obscures' or something like this

L79, 'the complex landscape' is vague – explain.

L80, delete 'the' after 'simulates' and provide a citation for the GEOtop model (e.g. Endrizzi et al.)

L83, insert 'with the inclusion of freeze-thaw' after 'improved performance', otherwise the sentence is a bit unclear.

L86, insert 'the' before 'global'

L87, Change 'The' to 'These

L90, Change 'were inadequate' to 'are lacking' which seems less of a personal attack

L93, Change 'Different from' to 'In contrast with'

L97, Delete 'the' after 'soil,'

L112, Delete 'the' after 'variations of'

L116, Change 'insufficient' to 'lacking' or something like this. This indicates that the

studies are lacking in number not in quality (which insufficient could imply).

L116, Insert 'an' before 'integrated'

L131, Start a new sentence after 1009 km2. Change 'it supplies' to '. The upper reaches supply'

L133, Insert 'the' after 'dominates'

L134, Insert comma after '(Wang et al., 2013)' as this is a coordinating conjunction connecting two independent clauses

L167, Delete 'which is' and insert comma after '2013'

L173, delete 'and' after T3

L196, delete 'of'

L197, Insert 'and' after 'glacier,'

L201, Should 'unit' be inserted after 'storage'??

L206, Please provide original citation for Horton-Strahler scheme not Yang papers.

L250, Is it very unusual to represent hydraulic permeability with a capital K. Usually this nomenclature refers to hydraulic conductivity. Recommend the authors use little k.

L260, The authors use the term heat capacity on L232 to refer to a volumetric heat capacity and on L260 to refer to a mass-based heat capacity. To be more consistent, the authors could refer to the mass-based heat capacity using its common term 'specific heat'

L262, insert 'vertical' after 1D to be clearer

L266, insert units after saturated soil hydraulic conductivity

L303, What is 'groundwater conductivity'? Hydraulic conductivity?

L353, 'value' should be plural

L360-361, this last sentence is not needed. The authors should not make statements like this. Allow the reader to come to their own conclusion on the suitability of the model.

L368, insert 'initiates' after 'thawing'

L374, Delete 'of'

L377, insert comma after 'seasons'

L497, Insert coma after 'season'

L500, Insert comma after 'low'

L509, Change 'lacked of' to 'did not consider' or something like that.

L511, delete 'the' before 'increased'

Section 5.3: The authors talk about uncertainty and how some processes can lead to overestimation of thaw. They should consider processes that can also lead to underestimation. For example, in warm permafrost environments, where permafrost coverage is discontinuous, and in complex terrain, lateral thawing can be very important and can accelerate thaw. Suggested studies are below. The heat transfer in the present study is only vertical and cannot accommodate lateral permafrost thaw. The authors should acknowledge this.

Noetzli et al. 2007. Three dimensional distribution and evolution of permafrost temperatures in idealized high-mountain topography. J. Geophys. Res.

Sjoberg et al. 2016. Thermal effects of groundwater flow through subarctic fens: A case study based on field observations and numerical modeling. Water Resour. Res.

Kurylyk et al. 2016. Influence of vertical and lateral heat transfer on permafrost thaw, peatland landscape transition, and groundwater flow. Water Resour. Res.

In this section on uncertainty, the Wu et al. (2016) paper deserves a citation as the modeling was similar and the paper was focused on uncertainty.

Wu et al. 2016. Constraining parameter uncertainty in simulations of water and heat dynamics in seasonally frozen soil using limited observed data. Water 8(2):64, doi:10.3390/w8020064

L580-581, 'on the high and cold plateau' is confusing and sounds odd

L588, insert 'sourced by' after 'mainly'

Figures are mostly very good. Figure 10 caption: Change 'at' to 'on' in both instances. Would it be possible to just plot the distribution of the land area that changed from permafrost to seasonally frozen ground?

---

## Author Comment (AC1) · 13 Mar 2017

**Reply to the Referee Comment by Anonymous Referee #1**

**General comments**

Gao et al. present a very impressive study that investigates how changing climate conditions influence hydrologic processes in a cold environment (particularly the Heihe basin in the Qinghai-Tibetan Plateau). Their model enhancements include algorithms for considering frozen ground processes. As such, they address how the hydrologic system responds to forcing (changing precipitation) but also to the changing system characteristics themselves (i.e. type and distribution of frozen ground). This study certainly has merit for publication in TC and has potential to be an important paper in this field. There are a few issues that I think should be considered prior to publication. Most of these are relatively minor, but addressing them all will likely warrant a major revision.

**Reply**: Thanks for providing very constructive and detailed comments on our manuscript. We have revised the manuscript and added more interpretation according to your suggestions.

**Major comments**

1. I think the introduction should be improved. (1) In the first few paragraphs, the authors go from regional to global and back to regional. I find this confusing. It would be preferable to start global and then narrow down to the Qinghai-Tibetan Plateau. It should be rewritten. (2) There are insufficient references, especially in the second half of the first paragraph. Many of those statements are not self-evident and should be backed up by more studies. I recommend the authors consider the already cited Walvoord and Kurylyk (2016, VZJ) review and references therein to back up these claims about how frozen soil and its evolution influence hydrological processes.

**Reply**: Thanks for your suggestions, we have rearranged the introduction part by starting from the global and then narrowing down to the Qinghai-Tibetan Plateau. And we cited "Walvoord and Kurylyk (2016, VZJ)" and other references to back up claims about how frozen soil and its evolution influence hydrological processes.

2. There are quite a few places where English issues occur. In general, the manuscript is in pretty good shape though. I do note a number of specific English corrections below in my minor comments.

**Reply**: Thanks for providing these detailed comments, we have corrected all the errors mentioned in your minor comments.

3. I'm curious if the authors have any citations to provide for the GBEHM model other than the Yang et al. 2015 paper published in a Chinese journal. The description in Section 3.1 is a bit brief, and there is no English source to go to for more information. One question: how are heads calculated in the aquifer since Darcy's Law drives GWSW

exchange?

**Reply**: Yang et al. (2015) was published in an English journal (Science China Earth Sciences), we also have another most recent paper by Gao et al. (2016) which introduced the GBEHM model. The GBEHM was developed on the basis of the geomorphology-based hydrological model (Yang et al., 1998 and 2002; Cong et al., 2009). These citations have been added in Section 3.1. The groundwater depth is assumed to be parallel to the bedrock. Changes in the groundwater depth are determined by the recharge from the unsaturated zone and exchange with the river.

The related citations:

Gao B., Qin Y., Wang YH, Yang DW, and Zheng YR: Modeling Ecohydrological Processes and Spatial Patterns in the Upper Heihe Basin in China, Forests, 7(1),10, doi:10.3390/f7010010, 2016.

Yang, D.W., Herath, S., and Musiake, K.: Development of a geomorphology-based hydrological model for large catchments, Annu. J. Hydraul. Eng., 42, 169-174, doi: 10.2208/prohe.42.169, 1998.

Yang, D.W., Herath, S., and Musiake, K.: A hillslope-based hydrological model using catchment area and width functions, Hydrol. Sci. J., 47, 49-65, doi: 10.1080/02626660209492907, 2002.

Cong Z T, Yang D W, Gao B, et al.: Hydrological trend analysis in the Yellow River basin using a distributed hydrological model, Water Resour Res, 45: W00A13, doi: 10.1029/2008WR006852, 2009

4. Equation 4. I'm a bit confused on the snowmelt equation, but I'm more used to frozen ground modeling. Why does the latent heat term only allow for ice to water transition and not snow to water? Is this ice referring to firn? This should be more explicit.

**Reply**: The parameterizations for snow are based on Jordan (1991). Snow is porous media and described by two constituents, the ice and liquid water. Water vapor phase is usually neglected. We added a sentence to describe this in the text as "The parametrization of snow is based on Jordan (1991), and each snow layer is described by two constituents, the ice and liquid water."

5. It is stated that Equations 4 and 8 are solved by the 'finite differential method'. Do the authors mean 'finite difference method'? If so, what method? Are the spatial derivatives solved as forward, backward or centered finite difference? Please state in the text. Is the time derivative implicit or explicit finite difference? Is there a Crank-Nicholson scheme employed? Just one or two more sentences will suffice.

**Reply**: It means finite difference method. Equations 4 and 8 are solved using implicit centered finite difference method and a Crank-Nicholson scheme is employed.

6. The authors would do well to include a figure of their modeling domain – at least their vertical discretization and boundary conditions. I find this lacking.

**Reply**: Thank you for this suggestion, we have added a figure (the following figure) to

show the model structure, vertical discretization and boundary conditions.

[Figure]

(a) Model structure                   (b)Vertical discretization

7. I realize this contradicts the statement above, but this is a bit long of a paper and there are many figures! I went through them and found them mostly valuable, however. I do think that Table 5 could be cut without much of a loss and Figure 14 could be as well. If the authors are in love with these, they could move them to an electronic supplement.

**Reply**: Thank you for this suggestions, we have cut Table 5 and move Figure 14 to the electronic supplement. We also modified the discussions about Table 5 and Figure 14 in the revised manuscript.

8. I'm confused at how the authors formed permafrost in 50 year spin up run (e.g. L297 and elsewhere). I've modeled even thin permafrost using 1000 year spin ups. Can the authors please describe this better? I wonder if this lack of permafrost formation time might partially cause some of the poor fits seen in Figure 2. Related to this, I wonder if the fits in Figure 2 in the shallow zone would look better at another time in the year (which the authors likely don't have data for). In other words, it could be the seasonal dynamics that are off, not the multi-decadal dynamics.

**Reply**: GBEHM saves the status variables (e.g., soil moisture, soil temperature, soil ice content, groundwater table, etc.) during the model calibration. The 50 year spin up run used the saved values as the initial conditions. So the actual spin up run was more than 50 years. We also carried out a 1000 year spin up run and compared the difference from the previous result (shown in the following figure). We found that the soil temperature is not so sensitive to the spin up time. It may be because that permafrost is shallow in the upper Heihe basin.

[Figure]

9. L330 and surrounding text, the authors suggest that RMSE decreases with increasing depth in Figure 3. It looks to me however (somewhat counter to what intuition would suggest) that the errors are increasing with depth (simulation – observation in Figure 3c goes to dark blue at the bottom indicating higher error and is white, indicating low error, higher up).

**Reply**: Thank you very much for this comments. We checked the calculation of RMSE and the plot of Figure 3c. It is confirmed that RMSE decreased with increasing depths though we have made small mistakes in calculation of RMSE for some depths. The problem is due to the coarse resolution of the color bar and errors for interpolation of the shaded counter in Figure 3c. We have also updated Figure 3c as follow.

[Figure]

10. The model results are impressive in Figure 4 (no small feat to get seasonal thaw and frost depths so well captured) and Figure 5 and Figure 6, so I commend the modelers on this.
**Reply**: Thank you for this comments.

11. L368. The authors should explain why the warming rate is higher in the shallower soils. It has to do with the surface signal arriving as a 'thermal breakthrough curve' that is retarded at greater depths due to the thermal inertia (sensible and latent heat storage) of the subsurface.
**Reply**: Thank you for this comment. We added explanation as "The larger warming trend in shallow soils is because the surface heat flux is retarded due to the thermal inertia when it penetrates to greater depths."

12. I don't understand the point of presenting some of the correlation information. It is Tautological to say that frost depth and thaw depth are correlated or inversely correlated to mean annual air temperature (L398-402). Along these lines, Table 5 is also not that useful as already stated. L463-470 is also a waste of space. You don't have to talk about correlation when you are using a physically-based model. Talk about physical processes!
**Reply**: We delete Table 5 and delete the discussions based on the correlation analysis in L463-470.

13. L429 and elsewhere, the authors talk about subsurface flow a bit confusingly. Is this

groundwater flow? Or is it groundwater flow plus lateral flow in the unsaturated zone? In reality all the precipitation typically makes its way to the river via subsurface flow paths as Hortonian overflow is very rare except in urbanized watersheds with impermeable pavement. They should be a bit clearer if they are talking about groundwater or what. L479 says that higher moisture increases conductivity and thus subsurface flow. True. But again, what flow? If they are talking about groundwater flow, they could have higher lateral transmissivity, but that effect wouldn't be captured in their model, I don't think.

**Reply**: It is groundwater flow plus lateral flow in the unsaturated zone. We modified the sentence as "In the freezing season, since there was no glacier melt and snow melt (see Table 4), runoff was mainly the subsurface flow (groundwater flow and lateral flow from the unsaturated zone)".

14. L600-601, is it possible that the increased liquid groundwater storage came from phase change of ice to water, rather than increased recharge? Couldn't the authors just directly determine the recharge from their modeling rather than making inferences based on groundwater storage?

**Reply**: This conclusion was based on the simulation of groundwater storage. Meanwhile, we recognized that ice in permafrost layer in the groundwater aquifer is quite limited based on the borehole observations by Wang et al. (2013). The GRACE data also showed that groundwater storage increased during the period of 2003~2008 in the upper Heihe basin (Cao et al., 2012).

**Minor comments**

1. L20, insert 'a' before 'regional'
**Reply**: we have revised this as suggested.

2. L47, Comma after 'soils'
**Reply**: we have revised this as suggested.

3. L52, delete 'balance' after water
**Reply**: we have revised this as suggested.

4. L59 and 507, The Walvoord et al. (2016) study (by the way there are no cited Walvoord papers with more than two authors) should really be Walvoord and Striegl 2007 or one of her related studies as these are the original field studies. e.g. Walvoord and Striegl 2007. Increasing groundwater to stream discharge from permafrost thawing in the Yukon River basin: Potential impacts on lateral export of carbon and nitrogen. Geophys. Res. Lett.
**Reply**: We have use "Walvoord and Striegl, 2007" instead of "Walvoord et al., 2016".

5. L60, The authors miss a very recent study in NE China that is certainly related to the present study Duan et al., 2017, Increasing winter baseflow in response to

permafrost thaw and precipitation regime shifts in northeastern China. Water, 9(1).
**Reply**: Thank you for this suggestion, we added citation of Duan et al., 2017 here.

6.  L61, The authors suggest a 'few studies' argue this point, but only provide 1 citation. Either provide more citations, use 'e.g.' in the citation, or reword this slightly.
**Reply**: We added one more citation here (Jin et al., 2009) and use 'e.g.' in the citation.

7.  L62, These studies typically included long term hydrological (not just meteorological) Data
**Reply**: We modified this sentence as" These studies used either in-situ observations in experimental catchments or long-term meteorological or hydrological observations"

8.  L63, Change 'might lose' to 'obscures' or something like this
**Reply**: we have revised this as suggested.

9.  L79, 'the complex landscape' is vague – explain.
**Reply**: We have changed this sentence to "but do not represent the complex topography and its effect on hydrological processes"

10.  L80, delete 'the' after 'simulates' and provide a citation for the GEOtop model (e.g.Endrizzi et al.)
**Reply**: we have revised this as suggested and added citation of GEOtop (Endrizzi et al., 2014).

11.  L83, insert 'with the inclusion of freeze-thaw' after 'improved performance', otherwise the sentence is a bit unclear.
**Reply**: we have revised this as suggested.

12.  L86, insert 'the' before 'global'
**Reply**: we have revised this as suggested.

13.  L87, Change 'The' to 'These'
**Reply**: we have revised this as suggested.

14.  L90, Change 'were inadequate' to 'are lacking' which seems less of a personal attack
**Reply**: we have revised this as suggested.

15. L93, Change 'Different from' to 'In contrast with'
**Reply**: we have revised this as suggested.

16. L97, Delete 'the' after 'soil,'
**Reply**: we have revised this as suggested.

17. L112, Delete 'the' after 'variations of'
**Reply**: we have revised this as suggested.

18. L116, Change 'insufficient' to 'lacking' or something like this. This indicates that the studies are lacking in number not in quality (which insufficient could imply).
**Reply**: we have revised this as suggested.

19. L116, Insert 'an' before 'integrated'
**Reply**: we have revised this as suggested.

20. L131, Start a new sentence after 1009 km2. Change 'it supplies' to '. The upper reaches supply'
**Reply**: we have revised this as suggested.

21. L133, Insert 'the' after 'dominates'
**Reply**: we have revised this as suggested.

22. L134, Insert comma after '(Wang et al., 2013)' as this is a coordinating conjunction connecting two independent clauses
**Reply**: we have revised this as suggested.

23. L167, Delete 'which is' and insert comma after '2013'
**Reply**: we have revised this as suggested.

24. L173, delete 'and' after T3
**Reply**: we have revised this as suggested.

25. L196, delete 'of'
**Reply**: we have revised this as suggested.

26. L197, Insert 'and' after 'glacier,'
**Reply**: we have revised this as suggested.

27. L201, Should 'unit' be inserted after 'storage'??
**Reply**: storage → storage unit

28. L206, Please provide original citation for Horton-Strahler scheme not Yang papers.
**Reply**: We have used original citation for Horton-Strahler scheme (Strahler, 1957) instead of Yang papers.

29. L250, Is it very unusual to represent hydraulic permeability with a capital K. Usually this nomenclature refers to hydraulic conductivity. Recommend the authors use little k.
**Reply**: we have revised this as suggested.

30.  L260, The authors use the term heat capacity on L232 to refer to a volumetric heat capacity and on L260 to refer to a mass-based heat capacity. To be more consistent, the authors could refer to the mass-based heat capacity using its common term 'specific heat'

**Reply**: we have revised this as suggested.

31. L262, insert 'vertical' after 1D to be clearer

**Reply**: we have revised this as suggested.

32. L266, insert units after saturated soil hydraulic conductivity

**Reply**: we have revised this as suggested.

33. L303, What is 'groundwater conductivity'? Hydraulic conductivity?

**Reply**: groundwater conductivity → groundwater hydraulic conductivity

34. L353, 'value' should be plural

**Reply**: we have revised this as suggested.

35.  L360-361, this last sentence is not needed. The authors should not make statements like this. Allow the reader to come to their own conclusion on the suitability of the model.

**Reply**: We have delete this sentence.

36. L368, insert 'initiates' after 'thawing'

**Reply**: we have revised this as suggested.

37. L374, Delete 'of'

**Reply**: we have revised this as suggested.

38. L377, insert comma after 'seasons'

**Reply**: we have revised this as suggested.

39. L497, Insert coma after 'season'

**Reply**: we have revised this as suggested.

40. L500, Insert comma after 'low'

**Reply**: we have revised this as suggested.

41. L509, Change 'lacked of' to 'did not consider' or something like that.

**Reply**: we have revised this as suggested.

42. L511, delete 'the' before 'increased'

**Reply**: we have revised this as suggested.

43. Section 5.3: The authors talk about uncertainty and how some processes can lead to overestimation of thaw. They should consider processes that can also lead to underestimation. For example, in warm permafrost environments, where permafrost coverage is discontinuous, and in complex terrain, lateral thawing can be very important and can accelerate thaw. Suggested studies are below. The heat transfer in the present study is only vertical and cannot accommodate lateral permafrost thaw. The authors should acknowledge this.
Noetzli et al. 2007. Three dimensional distribution and evolution of permafrost temperatures in idealized high-mountain topography. J. Geophys. Res.
Sjoberg et al. 2016. Thermal effects of groundwater flow through subarctic fens: A case study based on field observations and numerical modeling. Water Resour. Res.
Kurylyk et al. 2016. Influence of vertical and lateral heat transfer on permafrost thaw, peatland landscape transition, and groundwater flow. Water Resour. Res.i
    In this section on uncertainty, the Wu et al. (2016) paper deserves a citation as the modeling was similar and the paper was focused on uncertainty.
Wu et al. 2016. Constraining parameter uncertainty in simulations of water and heat dynamics in seasonally frozen soil using limited observed data. Water 8(2):64, doi:10.3390/w8020064
**Reply**: Thank you for this suggestions. We added two sentences in section 5.3 to discuss the processes that can lead to underestimation as "For discontinuous permafrost, lateral heat flux may increase thaw rate (Kurylyk et al., 2016; Sjöberg et al., 2016) and this effect is not considered in the present study. This may lead to underestimation in thaw rate of discontinuous permafrost, especially in spring." We have added these citations of "Sjoberg et al. 2016", "Kurylyk et al. 2016" and Wu et al. (2016).

44. L580-581, 'on the high and cold plateau' is confusing and sounds odd
**Reply**: We have modified this as "in the high plateau with cold climate"

45. L588, insert 'sourced by' after 'mainly'
**Reply**: We have revised this as suggested.

46. Figures are mostly very good. Figure 10 caption: Change 'at' to 'on' in both instances. Would it be possible to just plot the distribution of the land area that changed from permafrost to seasonally frozen ground?
**Reply**: We modified caption of Figure 10 as suggested. We added a figure to show the distribution of the land area that changed from permafrost to seasonally frozen ground (shown in the following figure).

[Figure]

---

## Referee Comment (RC2) · Anonymous Referee #2 · 7 Apr 2017

Review for <Change in frozen soils and its effect on regional hydrology in the upper Heihe Basin, the Northeast Qinghai-Tibetan Plateau> by Gao et al.

The authors took an ditributed hydrologial model to study the longterm(1961-2013) change of frozen soil and its influence on hydrology in the upper Heihe basin. In general, the manuscript is very interesting to me, and the conclusions are generally convincing. Based on my personal review, this paper can be accepted for TC after revision.

Comments: 1. The gridded forcing data are interpolated from very few observational stations. Please add the discussions for the uncertainty from model inputs (forcing data; particularly the precipitation, and solar radiation). 2. The tables can be reduced.

[Figure]

In my opinion, Table 2 can be merged into Figure 6. Tables 3 and 4 can be merged together. 3. Figure 7: how did you get the remote sensing estimated ET, please indicate the reference here.

---

## Author Comment (AC2) · 9 Apr 2017

Reply to the Referee Comment by Anonymous Referee #2

General comments

The authors took an distributed hydrological model to study the long term(1961-2013) change of frozen soil and its influence on hydrology in the upper Heihe basin. In general, the manuscript is very interesting to me, and the conclusions are generally convincing. Based on my personal review, this paper can be accepted for TC after revision.

Reply: Thanks for this positive comment, we will revise the manuscript by carefully following yours suggestions.

[Figure]

Major Comments

Comment 1. The gridded forcing data are interpolated from very few observational stations. Please add the discussions for the uncertainty from model inputs (forcing data; particularly the precipitation, and solar radiation).

Reply: The gridded forcing data was interpolated from the meteorological observations together with the outputs of a high resolution regional climate model. The interpolation method considered the precipitation–elevation relationship in the mountain region and has been validated carefully in the Heihe basin. The details of this method can be found in a recent publications by Wang et al. (2017) which has been added in the reference list of our manuscript (Wang et al. Spatial Interpolation of Daily Precipitation in a High Mountainous Watershed Based on Gauge Observations and a Regional Climate Model Simulation, Journal of Hydrometeorology, 18:845-862, 2017, doi: 10.1175/JHM-D-16-0089.1). We will add a discussion of the uncertainty from model inputs in section 5.3.

Comment 2. The tables can be reduced. In my opinion, Table 2 can be merged into Figure 6. Tables 3 and 4 can be merged together.

Reply: Thank you for this suggestions. We will merge Table 2 into Figure 6, and merge Table 3 and 4 into one table.

Comment 3. Figure 7: how did you get the remote sensing estimated ET, please indicate the reference here.

Reply: The remote sensing estimated ET was provided by Wu (2013) and the method for ET estimation was developed by Wu et al. (2012). We will add this information in the caption of Figure 7 "Comparison of the simulated and the remote sensing estimated actual evapotranspiration provided by Wu (2013) in the period of 2002~2012". The following references have been added in the manuscript.

Wu, B.F., Yan, N.N., Xiong, J., Bastiaanssen, W., Zhu, W.W., Stein, A.: Validation of ETWatch using field measurements at diverse landscapes: A case study in Hai Basin

of China. J. Hydrol., 436, 67-80, doi: 10.1016/j.jhydrol.2012.02.043, 2012.

Wu, B.F.: Monthly Evapotranspiration Datasets (2000–2012) with 1 km Spatial Resolution over the Heihe River Basin, Heihe Plan Science Data Center at Lanzhou, China, doi: 10.3972/heihe.115.2013.db, 2013.

---

## Editor Comment (EC1) · P.D. Morse (Editor) · 10 Apr 2017

Dear. Dr. Gao,

Manuscript tc-2016-289 entitled "Change in Frozen Soils and its Effect on Regional Hydrology in the Upper Heihe Basin, the Northeast Qinghai-Tibetan Plateau" which you submitted to The Cryosphere Discussions has been reviewed.

The referees found your paper to be interesting and suitable for publication in The Cryosphere, however they had some important concerns that require minor revisions of the manuscript. Their comments are helpful and self-explanatory.

A revised version of your manuscript that takes into account the comments of the ref-

erees will be reconsidered for publication.

Please note that submitting a revision of your manuscript does not guarantee eventual acceptance, and that your revision may be subject to re-review by the referees before a decision is rendered.

Thank you for submitting your manuscript to The Cryosphere Discussions, and I look forward to receiving your revision.

Regards,

Dr. Peter Morse

---

## Author Comment (AC3) · 25 Apr 2017

Thank you very much for your suggestions and comments. We have submitted the revised manuscript and author's response.
* * *